# Why do tree-based models still outperform deep learning on typical tabular data?

**Léo Grinsztajn**
Soda, Inria Saclay
leo.grinsztajn@inria.fr

**Edouard Oyallon**
MLIA, Sorbonne University

**Gaël Varoquaux**
Soda, Inria Saclay

## Abstract

While deep learning has enabled tremendous progress on text and image datasets, its superiority on tabular data is not clear. We contribute extensive benchmarks of standard and novel deep learning methods as well as tree-based models such as XGBoost and Random Forests, across a large number of datasets and hyperparameter combinations. We define a standard set of 45 datasets from varied domains with clear characteristics of tabular data and a benchmarking methodology accounting for both fitting models and finding good hyperparameters. Results show that tree-based models remain state-of-the-art on medium-sized data (∼10K samples) even without accounting for their superior speed. To understand this gap, we conduct an empirical investigation into the differing inductive biases of tree-based models and neural networks. This leads to a series of challenges which should guide researchers aiming to build tabular-specific neural network: **1.** be robust to uninformative features, **2.** preserve the orientation of the data, and **3.** be able to easily learn irregular functions. To stimulate research on tabular architectures, we contribute a standard benchmark and raw data for baselines: every point of a 20 000 compute hours hyperparameter search for each learner.

## 1  Introduction

Deep learning has enabled tremendous progress for learning on image, language, or even audio datasets. On tabular data, however, the picture is muddier and ensemble models based on decision trees like XGBoost remain the go-to tool for most practitioners [Sta] and data science competitions [Kossen et al., 2021]. Indeed deep learning architectures have been crafted to create inductive biases matching invariances and spatial dependencies of the data. Finding corresponding invariances is hard in tabular data, made of heterogeneous features, small sample sizes, extreme values.

Creating tabular-specific deep learning architectures is a very active area of research (see section 2). One motivation is that tree-based models are not differentiable, and thus cannot be easily composed and jointly trained with other deep learning blocks. Most tabular deep learning publications claim to beat or match tree-based models, but their claims have been put into question: a simple Resnet seems to be competitive with some of these new models [Gorishniy et al., 2021], and most of these methods seem to fail on new datasets [Shwartz-Ziv and Armon, 2021]. Indeed, the lack of an established benchmark for tabular data learning provides additional degrees of freedom to researchers when evaluating their method. Furthermore, most tabular datasets available online are small compared to benchmarks in other machine learning subdomains, such as ImageNet [Ima], making evaluation noisier. These issues add up to other sources of unreplicability across machine learning, such as unequal hyperparameters tuning efforts [Lipton and Steinhardt, 2019] or failure to account for statistical uncertainty in benchmarks [Bouthillier et al., 2021]. To alleviate these concerns, we contribute a tabular data benchmark with a precise methodology for datasets inclusion and hyperparameter tuning. This enables us to evaluate recent deep learning models which have

36th Conference on Neural Information Processing Systems (NeurIPS 2022) Track on Datasets and Benchmarks.

not yet been independently evaluated, and to show that tree-based models remain state-of-the-art on medium-sized tabular datasets, even without accounting for the slower training of deep learning algorithms. Furthermore, we show that this performance gap is not mostly due to categorical features, and does not disappear after tuning hyperparameters.

Impressed by the superiority of tree-based models on tabular data, we strive to understand which *inductive biases* make them well-suited for these data. By transforming tabular datasets to modify the performances of different models, we uncover differing biases of tree-based models and deep learning algorithms which partly explain their different performances: neural networks struggle to learn irregular patterns of the target function, and their rotation invariance hurt their performance, in particular when handling the numerous uninformative features present in tabular data.

Our contributions are as follow: **1.** We create a new benchmark for tabular data, with a precise methodology for choosing and preprocessing a large number of representative datasets. We share these datasets through OpenML [Vanschoren et al., 2014], which makes them easy to use. **2.** We extensively compare deep learning models and tree-based models on generic tabular datasets in multiple settings, accounting for the cost of choosing hyperparameters. We also share the raw results of our random searches, which will enable researchers to cheaply test new algorithms for a fixed hyperparameter optimization budget. **3.** We investigate empirically why tree-based models outperform deep learning, by finding data transformations which narrow or widen their performance gap. This highlights desirable biases for tabular data learning, which we hope will help other researchers to successfully build deep learning models for tabular data.

In Sec. 2 we cover related work. Sec. 3 gives a short description of our benchmark methodology, including datasets, data processing, and hyper-parameter tuning. Then, Sec. 4 shows our raw results on deep learning and tree-based models after an extensive random search. Finally, Sec. 5 provides the results of an empirical study which exhibit desirable implicit biases of tabular datasets.[1]

## 2   Related work

**Deep learning for tabular data**   As described by Borisov et al. [2021] in their review of the field, there have been various attempts to adapt deep learning to tabular data: data encoding techniques to make tabular data better suited for deep learning [Hancock and Khoshgoftaar, 2020, Yoon et al., 2020], "hybrid methods" to benefit from the flexibility of neural networks while keeping the inductive biases of other algorithms like tree-based models [Lay et al., 2018, Popov et al., 2020, Abutbul et al., 2020, Hehn et al., 2019, Tanno et al., 2019, Chen, 2020, Kontschieder et al., 2015, Rodriguez et al., 2019, Popov et al., 2020, Lay et al., 2018] or Factorization Machines Guo et al. [2017], tabular-specific transformers architectures Somepalli et al. [2021], Kossen et al. [2021], Arik and Pfister [2019], Huang et al. [2020], and various regularization techniques to adapt classical architectures to tabular data [Lounici et al., 2021, Shavitt and Segal, 2018, Kadra et al., 2021a, Fiedler, 2021]. In this paper, we focus on architectures directly inspired by classic deep learning models, in particular Transformers and Multi-Layer-Perceptrons (MLPs).

**Comparisons between neural networks and tree-based models**   The most comprehensive comparisons of machine learning algorithms have been published before the advent of new deep learning methods [Caruana and Niculescu-Mizil, 2006, Fernández-Delgado et al., 2014], or on specific problems [Sakr et al., 2017, Korotcov et al., 2017, Uddin et al., 2019]. Recently, Shwartz-Ziv and Armon [2021] evaluated modern tabular-specific deep learning methods, but their goal was more to reveal that "New deep learning architectures fail to generalize to new datasets" than to create a comprehensive benchmark. Borisov et al. [2022] benchmarked recent algorithms in their review of deep learning for tabular data, but only on 3 datasets, and "highlight[ed] the need for unified benchmarks" for tabular data. Most papers introducing a new architecture for tabular data benchmark various algorithms, but with a highly variable evaluation methodology, a small number of datasets, and the evaluation can be biased toward the authors' model [Shwartz-Ziv and Armon, 2021]. The paper closest to our work is Gorishniy et al. [2021], benchmarking novel algorithms, on 11 tabular datasets. We provide a more comprehensive benchmark, with 45 datasets, split across different settings (medium-sized

---

[1]Compared to our initial submission, the final version of this paper includes a simple decision tree as a baseline. In addition, it displays updated figures with minor bug fixes which do not affect our conclusions.

/ large-size, with/without categorical features), accounting for the hyperparameter tuning cost, to establish a standard benchmark.

**No standard benchmark for tabular data**  Unlike other machine learning subfields such as computer vision [Ima] or NLP [Wang et al., 2020], there are no standard benchmarks for tabular data. There exist generic machine learning benchmarks, but, to the our knowledge, none are specific to tabular data. For instance, OpenML benchmarks CC-18, CC-100, [Bischl et al., 2021] and AutoML Benchmark [Gijsbers et al., 2019] contain tabular data, but also include images and artificial datasets, which may explain why they have not been used in tabular deep learning papers. In A.6, we compare in more depth our benchmark to these previous ones.

**Understanding the difference between neural networks and tree-based models**  To our knowledge, this is the first empirical investigation of *why* tree-based models outperform neural networks on tabular data. Some speculative explanations, however, have been offered [Klambauer et al., 2017, Borisov et al., 2021]. Kadra et al. [2021a] claims that searching across 13 regularization techniques for MLPs to find a dataset-specific combination gives state-of-the-art performances. This provides a partial explanation: MLPs are expressive enough for tabular data but may suffer from a lack of proper regularization.

## 3 A benchmark for tabular learning

### 3.1 45 reference tabular datasets

For our benchmark, we compiled 45 tabular datasets from various domains provided mainly by OpenML, listed in A.1 and selected via the following criteria:

**Heterogeneous columns.**  Columns should correspond to features of different nature. This excludes images or signal datasets where each column corresponds to the same signal on different sensors.
**Not high dimensional.**  We only keep datasets with a $d/n$ ratio below 1/10, and with $d$ below 500.
**Undocumented datasets**  We remove datasets where too little information is available. We did keep datasets with hidden column names if it was clear that the features were heterogeneous.
**I.I.D. data.**  We remove stream-like datasets or time series.
**Real-world data.**  We remove artificial datasets but keep some simulated datasets. The difference is subtle, but we try to keep simulated datasets if learning these datasets are of practical importance (like the Higgs dataset), and not just a toy example to test specific model capabilities.
**Not too small.**  We remove datasets with too few features ($< 4$) and too few samples ($< 3\,000$). For benchmarks on numerical features only, we remove categorical features before checking if enough features and samples are remaining.
**Not too easy.**  We remove datasets which are too easy. Specifically, we remove a dataset if a simple model (max of a single tree and a regression, logistic or OLS) reaches a score whose relative difference with the score of both a default Resnet (from Gorishniy et al. [2021]) and a default HistGradientBoosting model (from scikit learn) is below 5%. Other benchmarks use different metrics to remove too easy datasets, like removing datasets perfectly separated by a single decision classifier [Bischl et al., 2021], but this ignores varying Bayes rate across datasets. As tree ensembles are superior to simple trees and logistic regresison [Fernández-Delgado et al., 2014], a close score for the simple and powerful models suggests that we are already close to the best achievable score.
**Not deterministic.**  We remove datasets where the target is a deterministic function of the data. This mostly means removing datasets on games like poker and chess. Indeed, we believe that these datasets are very different from most real-world tabular datasets, and should be studied separately.

### 3.2 Removing side issues

To keep learning tasks as homogeneous as possible and focus on challenges specific to tabular data, we exclude subproblems which would deserve their own analysis:

**Medium-sized training set**  We truncate the training set to 10,000 samples for bigger datasets. This allows us to investigate the medium-sized dataset regime. We study the large-sized (50,000) regime, for which fewer datasets matching our criteria are available, in A.2.
**No missing data**  We remove all missing data from the datasets. Indeed, there are numerous techniques for handling missing data both for tree-based models and neural networks, with varying

performances [Perez-Lebel et al., 2022]. In practice, we first remove columns containing many missing data, then all rows containing at least one missing entry.

**Balanced classes** For classification, the target is binarised if there are several classes, by taking the two most numerous classes, and we keep half of samples in each class.

**Low cardinality categorical features** We remove categorical features with more than 20 items.

**High cardinality numerical features** We remove numerical features with less than 10 unique values. Numerical features with 2 unique values are converted to categorical features.

### 3.3 A procedure to benchmark models with hyperparameter selection

Hyperparameter tuning leads to uncontrolled variance on a benchmark [Bouthillier et al., 2021], especially with a small budget of model evaluations. We design a benchmarking procedure that jointly samples the variance of hyperparameter tuning and explores increasingly high budgets of model evaluations. It relies on random searches for hyper-parameter tuning [Bergstra et al., 2013]. We use hyperparameter search spaces from the Hyperopt-Sklearn [Komer et al., 2014] when available, from the original paper when possible, and from Gorishniy et al. [2021] for MLP, Resnet and XGBoost (see A.3). We run a random search of $\approx 400$ iterations per dataset, on CPU for tree-based models and GPU for neural networks (more details in A.3).

To study performance as a function of the number $n$ of random search iterations, we compute the best hyperparameter combination on the validation set on these $n$ iterations (for each model and dataset), and evaluate it on the test set. We do this 15 times while shuffling the random search order at each time. This gives us bootstrap-like estimates of the expected test score of the best (on the validation set) model after each number of random search iterations. In addition, we always start the random searches with the default hyperparameters of each model. In A.7, we show that using Bayesian optimization instead of random search does not seem to change our results.

**Resuable code and benchmark raw data** The code used for all the experiments and comparisons is available at https://github.com/LeoGrin/tabular-benchmark. To help researchers to cheaply add their own algorithms to the results, we also share at the same link a data table containing results for all iterations of our 20,000 compute-hour random searches.

### 3.4 Aggregating results across datasets

We use the test set accuracy (classification) and R2 score (regression) to measure model performance. To aggregate results across datasets of varying difficulty, we use a metric similar to the distance to the minimum (or average distance to the minimum –ADTM– when averaged across datasets), used in Feurer et al. [2021] and introduced in Wistuba et al. [2015]. This metric consists in normalizing each test accuracy between 0 and 1 via an affine renormalization between the top-performing and worse-performing models.[2] Instead of the worse-performing model, we use models achieving the 10% (classification) or 50% (regression) test error quantile. Indeed, the worse scores are achieved by outlier models and are not representative of the difficulty of the dataset. For regression tasks, we clip all negative scores (i.e below 50% scores) to 0 to reduce the influence of very low scores.

### 3.5 Data preparation

We strive for as little manual preprocessing as possible, applying only the following transformations:

**Gaussianized features** For neural network training, the features are Gaussianized with Scikit-learn's `QuantileTransformer`.

**Transformed regression targets** In regression settings, target variables are log-transformed when their distributions are heavy-tailed (e.g house prices, see A.1). In addition, we add as an hyperparameter the possibility to Gaussienize the target variable for model fit, and transform it back for evaluation (via ScikitLearn's TransformedTargetRegressor and QuantileTransformer).

**OneHotEncoder** For models which do not handle categorical variables natively, we encode categorical features using ScikitLearn's OneHotEncoder.

---

[2]This method is also close to the method used by Caruana and Niculescu-Mizil [2006], the difference being that the latter uses an artificial baseline (predicting the most common class) as the zero score.

# 4 Tree-based models still outperform deep learning on tabular data.

## 4.1 Models benchmarked

For tree-based models, we choose 3 state-of-the-art models used by practitioners: Scikit Learn's RandomForest, GradientBoostingTrees (GBTs) (or HistGradientBoostingTrees when using categorical features), and XGBoost [Chen and Guestrin, 2016]. We benchmark the following deep models:

**MLP** : a classical MLP from Gorishniy et al. [2021]. The only improvement beyond a simple MLP is using Pytorch's `ReduceOnPlateau` learning rate scheduler.

**Resnet** : as in Gorishniy et al. [2021], similar to **MLP** with dropout, batch/layer normalization, and skip connections.

**FT_Transformer** : a simple Transformer model combined with a module embedding categorical and numerical features, created in Gorishniy et al. [2021]. We choose this model because it was benchmarked in a convincing way against tree-based models and other tabular-specific models. It can thus be considered a "best case" for Deep learning models on tabular data.

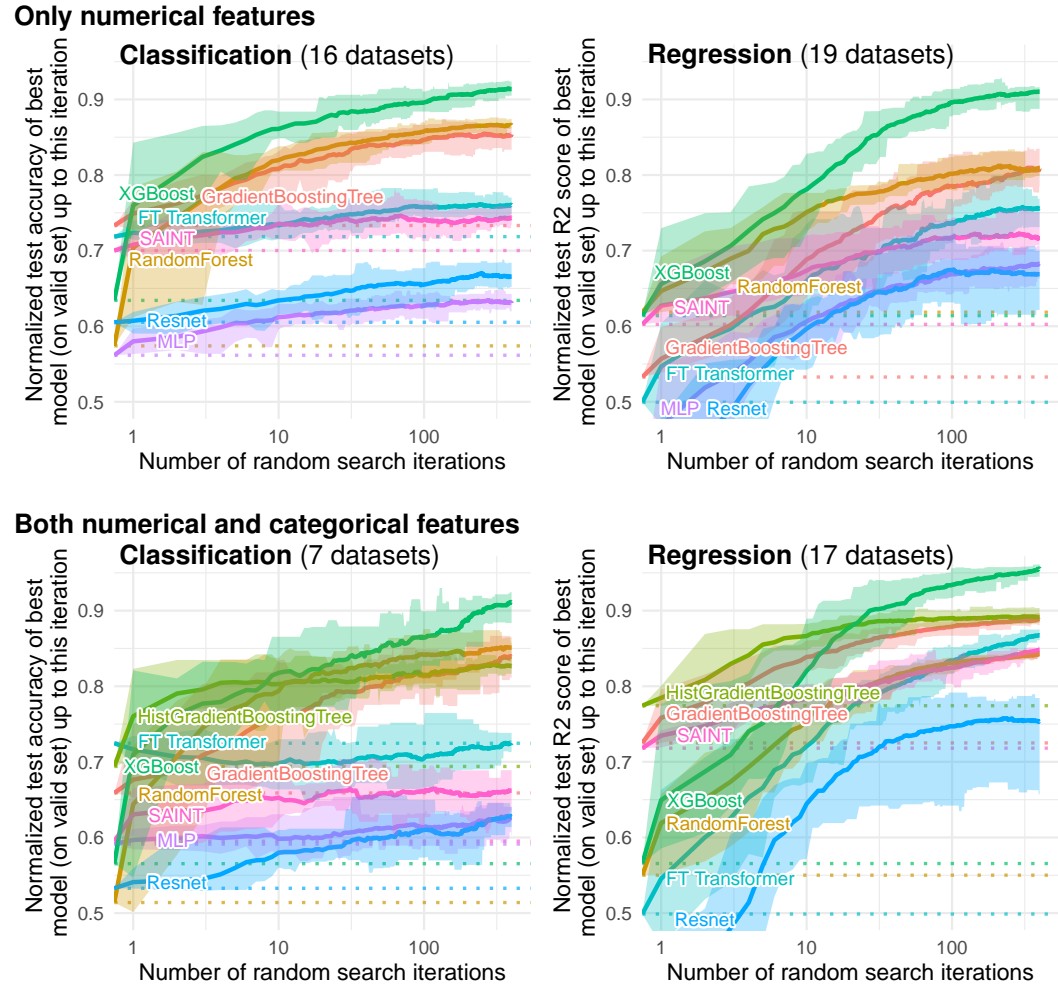

Figure 1: **Benchmark on medium-sized datasets**, top only numerical features; bottom: all features. Dotted lines correspond to the score of the default hyperparameters, which is also the first random search iteration. Each value corresponds to the test score of the best model (on the validation set) after a specific number of random search iterations, averaged on 15 shuffles of the random search order. The ribbon corresponds to minimum and maximum scores on these 15 shuffles.

**SAINT** : a Transformer model with an embedding module and an inter-samples attention mechanism, proposed in Somepalli et al. [2021]. We include this model because it was the best performing deep model in Borisov et al. [2021], and to investigate the impact of inter-sample attention, which performs well on tabular data according to Kossen et al. [2022].

## 4.2   Results

Fig. 1 give benchmark results for different types of datasets (appendix A.2 gives results as a function of computation *time*). We emphasize that the variance quantification in these figures should be interpreted carefully, as it is made by shuffling the order of a same random search: for a large number of random search iterations, it may not represent the actual variance after this number of step.

**Tuning hyperparameters does not make neural networks state-of-the-art**   Tree-based models are superior for every random search budget, and the performance gap stays wide even after a large number of random search iterations. This does not take into account that each random search iteration is generally slower for neural networks than for tree-based models (see A.2).

**Categorical variables are not the main weakness of neural networks**   Categorical variables are often seen as a major problem for using neural networks on tabular data [Borisov et al., 2021]. Our results on numerical variables only do reveal a narrower gap between tree-based models and neural networks than including categorical variables. Still, most of this gap subsists when learning on numerical features only.

## 5   Empirical investigation: *why* do tree-based models still outperform deep learning on tabular data?

### 5.1   Methodology: uncovering inductive biases

We have seen in Sec. 4.2 that tree-based models beat neural networks across a wide range of hyperparameter choices. This hints to inherent properties of these models which explains their performances on tabular data. Indeed, the best methods on tabular data share two attributes: they are **ensemble methods**, bagging (Random Forest) or boosting (XGBoost, GBTs), and the weak learner used in these ensembles is a **decision tree**. The decisive point seems to be the tree aspect: other boosting and bagging methods with different weak learners exist but are not commonly used for tabular data. In this section, we try to understand the *inductive biases* of decision trees that make them well-suited for tabular data, and how they differ from the inductive biases of neural networks. This is equivalent to saying the reverse: which features of tabular data make this type of data easy to learn with tree-based methods yet more difficult with a neural network?

To this aim, we apply various transformations to tabular datasets which either narrow or widen the generalization performance gap between neural networks and tree-based models, and thus help us emphasize their different inductive biases. For the sake of simplicity, we restrict our analysis to numerical variables and classification tasks on medium-sized datasets. Results are presented aggregated across datasets, and dataset-specific results are available in A.4, along with additional details on our experiments.

### 5.2   Finding 1: Neural networks are biased to overly smooth solutions

We transform each *train* set by smoothing the output with a Gaussian Kernel smoother for varying length-scale values of the kernel (more details are available in A.4). This effectively prevents models from learning irregular patterns of the target function. Fig. 2 shows model performance as a function of the length-scale of the smoothing kernel. For small lengthscales, smoothing the target function on the train set decreases markedly the accuracy of tree-based models, but barely impacts that of neural networks.

Such results suggest that the target functions in our datasets are not smooth, and that neural networks struggle to fit these irregular functions compared to tree-based models. This is in line with Rahaman et al. [2019], which finds that neural networks are biased toward low-frequency functions. Models based on decision trees, which learn piece-wise constant functions, do not exhibit such a bias. Our

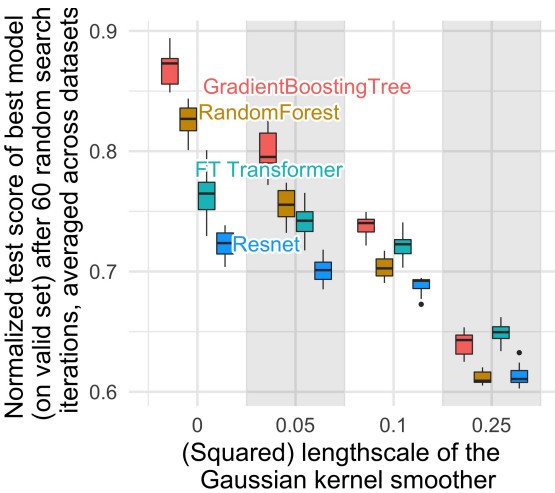

Figure 2: **Normalized test accuracy of different models for varying smoothing of the target function on the train set**. We smooth the target function through a Gaussian Kernel smoother, whose covariance matrix is the data covariance, multiplied by the (squared) lengthscale of the Gaussian kernel smoother. A lengthscale of 0 corresponds to no smoothing (the original data). All features have been Gaussienized before the smoothing through ScikitLearn's QuantileTransformer. The boxplots represent the distribution of normalized accuracies across 15 re-orderings of the random search.

findings do not contradict papers claiming benefits from regularization for tabular data [Shavitt and Segal, 2018, Borisov et al., 2021, Kadra et al., 2021b, Lounici et al., 2021], as adequate regularization and careful optimization may allow neural networks to learn irregular patterns. In A.4, we show some examples of non-smooth patterns which neural networks fail to learn, both in toy and real-world settings.

Note also that our observation could also explain the benefits of the ExU activation used in the Neural-GAM paper [Agarwal et al., 2021], and of the embeddings used in Gorishniy et al. [2022]: the periodic embedding might help the model to learn the high-frequency part of the target function, and the target-aware binning might make the target function smoother.

### 5.3   Finding 2: Uninformative features affect more MLP-like neural networks

**Tabular datasets contain many uninformative features**    For each dataset, we drop an increasingly large fraction of features, according to feature importance (ranked by a Random Forest). Fig. 3 shows that the classification accuracy of a GBT is not much affected by removing up to half of the features.

Furthermore, the test accuracy of a GBT trained on the removed features (i.e the features below a certain feature importance threshold) is very low up to 20% of features removed, and quite low until 50%, which suggests that most of these features are uninformative, and not solely redundant.

**MLP-like architectures are not robust to uninformative features**    In the two experiments shown in Fig. 4, we can see that *removing* uninformative features (4a) reduces the performance gap between MLPs (Resnet) and the other models (FT Transformers and tree-based models), while *adding* uninformative features widens the gap. This shows that MLPs are less robust to uninformative features, and, given the frequency of such features in tabular datasets, partly explain the results from Sec. 4.2.

In Fig. 4a, we also remove informative features as we remove a larger fraction of features. Our reasoning, which is backed by 4b, is that the decrease in accuracy due to the removal of these features is compensated by the removal of uninformative features, which is more helpful for MLPs than for other models (we also remove redundant features at the same time, which should not impact our models)

### 5.4   Finding 3: Data are non invariant by rotation, so should be learning procedures

Why are MLPs much more hindered by uninformative features, compared to other models? One answer is that this learner is rotationally invariant in the sense of Ng [2004]: the learning procedure which learns an MLP on a training set and evaluate it on a testing set is unchanged when applying a rotation (unitary matrix) to the features on both the training and testing set. On the contrary, tree-based models are not rotationally invariant, as they attend to each feature separately, and neither are FT Transformers, because of the initial FT Tokenizer, which implements a pointwise operation. A

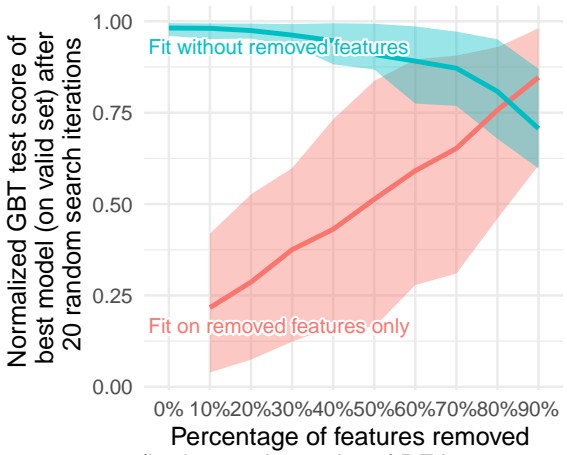

Figure 3: **Test accuracy of a GBT for varying proportions of removed features**, on our classification benchmark on numerical features. Features are removed in increasing order of feature importance (computed with a Random Forest), and the two lines correspond to the accuracy using the (most important) kept features (blue) or the (least important) removed features (red). A score of 1.0 corresponds to the best score across all models and hyperparameters on each dataset, and 0.0 correspond to random chance. These scores are averaged across 30 random search orders, and the ribbons correspond to the 80% interval among the different datasets.

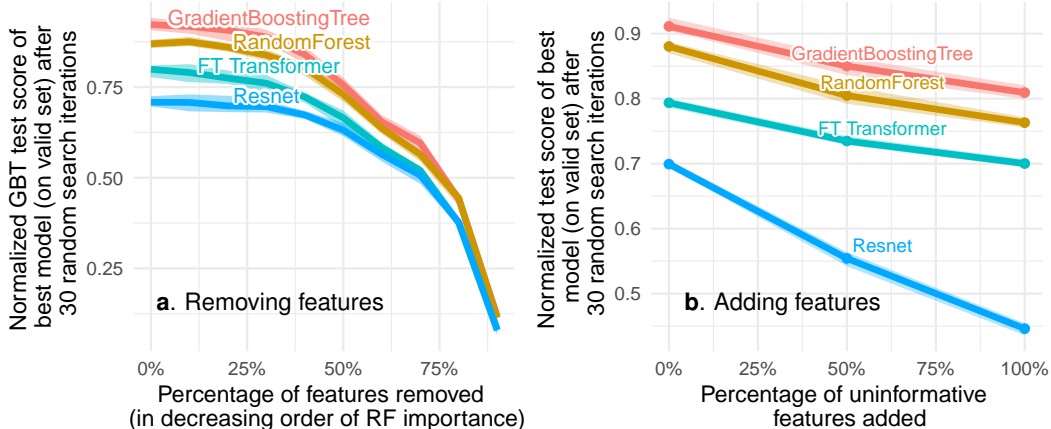

Figure 4: **Test accuracy changes when removing (a) or adding (b) uninformative features**. Features are removed in increasing order of feature importance (computed with a Random Forest). Added features are sampled from standard Gaussians uncorrelated with the target and with other features. Scores are averaged across datasets, and the ribbons correspond to the minimum and maximum score among the 30 different random search reorders (starting with the default models).

theoretical link between this concept and uninformative features is provided by Ng [2004], which shows that any rotationallly invariant learning procedure has a worst-case sample complexity that grows at least linearly in the number of irrelevant features. Intuitively, to remove uninformative features, a rotationaly invariant algorithm has to first find the original orientation of the features, and then select the least informative ones: the information contained in the orientation of the data is lost.

Fig. 5a, which shows the change in test accuracy when randomly rotating our datasets, confirms that only Resnets are rotationally invariant. More striking, random rotations reverse the performance order: neural networks are now above tree-based models and Resnets above FT Transformers. This suggests that rotation invariance is not desirable: similarly to vision [**?**], there is a natural basis (here, the original basis) which encodes best data-biases, and which can not be recovered by models invariant to rotations which potentially mixes features with very different statistical properties. Indeed, features of a tabular data typically carry meanings individually, as expressed by column names: age, weight. The link with uninformative features is apparent in 5b: removing the least important half of the features in each dataset (before rotating), drops the performance of all models except Resnets, but the decrease is less significant than when using all features.

Our findings shed light on the results of Somepalli et al. [2021] and Gorishniy et al. [2022], which add an embedding layer, even for numerical features, before MLP or Transformer models. Indeed,

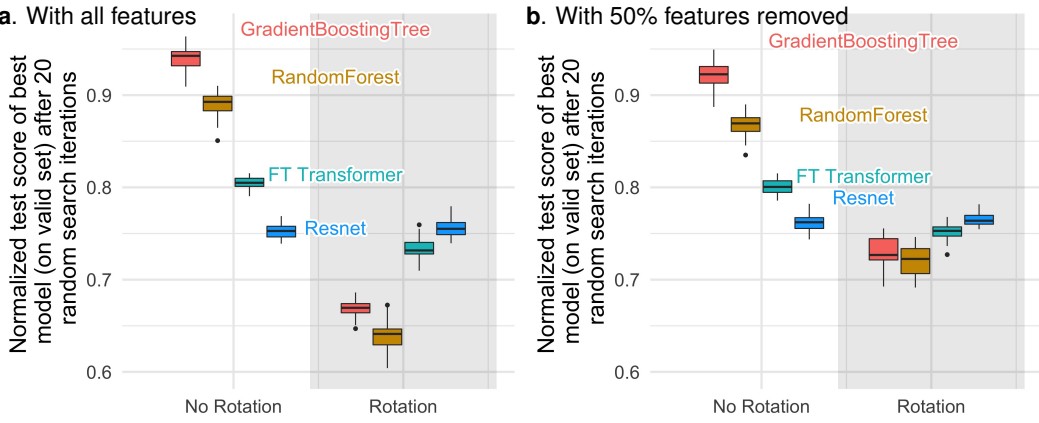

Figure 5: **Normalized test accuracy of different models when randomly rotating our datasets**. Here, the classification benchmark on numerical features was used. All features are Gaussianized before the random rotations. The scores are averaged across datasets, and the boxes depict the distribution across random search shuffles. Right: the features are removed before data rotation.

this layer breaks rotation invariance. The fact that very different types of embedding seem to improve performance suggests that the sheer presence of an embedding which breaks the invariance is a key part of these improvements. We note that a promising avenue for further research would be to find other ways to break rotation invariance which might be less computationally costly than embeddings.

# 6 Discussion and conclusion

**Limitation** Our study leaves open many questions for future work, such as: which other inductive biases of tree-based models explain their performances on tabular data? Our benchmarks could be extended in numerous ways:

- Similar analysis for different settings, such as small datasets, or very large datasets.
- Comparing the same algorithms on a new task, multi-class classification, which is a common task for tabular datasets.
- Investigating different metrics, especially metrics evaluating the probabilistic predictions on classification tasks [as in Caruana and Niculescu-Mizil, 2006].
- Study how both tree-based models and neural networks cope with specific challenges such as missing data or high-cardinality categorical features, thus extending to neural networks prior empirical work [Cerda et al., 2018, Cerda and Varoquaux, 2020, Perez-Lebel et al., 2022].

Another interesting path for future work would be to study the specific benefits of deep learning brings over tree-based models, for instance by studying the usefulness of the embeddings learnt by Neural networks for downstream tasks.

**Conclusion** While each publication on learning architectures for tabular data comes to different results using a different benchmarking methodology, our systematic benchmark, going beyond the specificities of a handful of datasets and accounting for hyper-parameter choice, reveals clear trends. On such data, tree-based models more easily yield good predictions, with much less computational cost. This superiority is explained by specific features of tabular data: irregular patterns in the target function, uninformative features, and non rotationally-invariant data where linear combinations of features misrepresent the information. Beyond these conclusions, our benchmark is reusable, allowing researchers to use our methodology and datasets for new architectures, and to easily compare them to those we explored via the shared benchmark raw results. We hope that this benchmark will stimulate tabular deep-learning research and foster more thorough empirical evaluation of contributions.

## Acknowledgments and Disclosure of Funding

GV and LG acknowledge support in part by the French Agence Nationale de la Recherche under Grant ANR-20-CHIA-0026 (LearnI). EO was supported by the Project ANR-21-CE23-0030 ADONIS and EMERG-ADONIS from Alliance SU.

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
