# A  Appendix

## A.1  Datasets used

We describe below all datasets used in our benchmarks, along with the link to the original dataset, as well as a new OpenML link to the transformed datasets used for our benchmarks. All datasets considered for the benchmarks, as well as the reason for their selection or their exclusion, are available at this link: `https://docs.google.com/spreadsheets/d/159UsoK3q2x-wXKoYEY-zHlZhiIcDgjCFbDW69QMmUnk/edit?usp=sharing` or on the Github repo; however, the authoritative list of datasets used for the present study and its figures is the list below. Instructions on how to use these datasets to benchmark your own algorithms are available at A.10.

### A.1.1  Numerical classification

OpenML benchmark: `https://www.openml.org/search?type=benchmark&study_type=task&sort=tasks_included&id=337`

| dataset_name | n_samples | n_features | original_link | new_link |
|---|---|---|---|---|
| electricity | 38474.0 | 7.0 | https://www.openml.org/d/151 | https://www.openml.org/d/44120 |
| covertype | 566602.0 | 10.0 | https://www.openml.org/d/293 | https://www.openml.org/d/44121 |
| pol | 10082.0 | 26.0 | https://www.openml.org/d/722 | https://www.openml.org/d/44122 |
| house_16H | 13488.0 | 16.0 | https://www.openml.org/d/821 | https://www.openml.org/d/44123 |
| MagicTelescope | 13376.0 | 10.0 | https://www.openml.org/d/1120 | https://www.openml.org/d/44125 |
| bank-marketing | 10578.0 | 7.0 | https://www.openml.org/d/1461 | https://www.openml.org/d/44126 |
| Bioresponse | 3434.0 | 419.0 | https://www.openml.org/d/4134 | https://www.openml.org/d/45019 |
| MiniBooNE | 72998.0 | 50.0 | https://www.openml.org/d/41150 | https://www.openml.org/d/44128 |
| default-of-credit-card-clients | 13272.0 | 20.0 | https://www.openml.org/d/42477 | https://www.openml.org/d/45020 |
| Higgs | 940160.0 | 24.0 | https://www.openml.org/d/42769 | https://www.openml.org/d/44129 |
| eye_movements | 7608.0 | 20.0 | https://www.openml.org/d/1044 | https://www.openml.org/d/44130 |
| Diabetes130US | 71090.0 | 7.0 | https://www.openml.org/d/4541 | https://www.openml.org/d/45022 |
| jannis | 57580.0 | 54.0 | https://www.openml.org/d/41168 | https://www.openml.org/d/45021 |
| credit | 16714.0 | 10.0 | "https://www.kaggle.com/c/GiveMeSomeCredit/data?select=cs-training.csv" | https://www.openml.org/d/44089 |
| california | 20634.0 | 8.0 | "https://www.dcc.fc.up.pt/ltorgo/Regression/cal_housing.html" | https://www.openml.org/d/45028 |

### A.1.2  Numerical regression

OpenML benchmark: `https://www.openml.org/search?type=benchmark&study_type=task&sort=tasks_included&id=336`

| dataset_name | n_samples | n_features | original_link | new_link |
|---|---|---|---|---|
| cpu_act | 8192.0 | 21.0 | https://www.openml.org/d/197 | https://www.openml.org/d/44132 |
| pol | 15000.0 | 26.0 | https://www.openml.org/d/201 | https://www.openml.org/d/44133 |
| elevators | 16599.0 | 16.0 | https://www.openml.org/d/216 | https://www.openml.org/d/44134 |
| wine_quality | 6497.0 | 11.0 | https://www.openml.org/d/287 | https://www.openml.org/d/44136 |
| Ailerons | 13750.0 | 33.0 | https://www.openml.org/d/296 | https://www.openml.org/d/44137 |
| yprop_4_1 | 8885.0 | 42.0 | https://www.openml.org/d/416 | https://www.openml.org/d/45032 |
| houses | 20640.0 | 8.0 | https://www.openml.org/d/537 | https://www.openml.org/d/44138 |
| house_16H | 22784.0 | 16.0 | https://www.openml.org/d/574 | https://www.openml.org/d/44139 |
| delays_zurich_transport | 5465575.0 | 9.0 | https://www.openml.org/d/40753 | https://www.openml.org/d/45034 |
| diamonds | 53940.0 | 6.0 | https://www.openml.org/d/42225 | https://www.openml.org/d/44140 |
| Brazilian_houses | 10692.0 | 8.0 | https://www.openml.org/d/42688 | https://www.openml.org/d/44141 |
| Bike_Sharing_Demand | 17379.0 | 6.0 | https://www.openml.org/d/42712 | https://www.openml.org/d/44142 |
| nyc-taxi-green-dec-2016 | 581835.0 | 9.0 | https://www.openml.org/d/42729 | https://www.openml.org/d/44143 |
| house_sales | 21613.0 | 15.0 | https://www.openml.org/d/42731 | https://www.openml.org/d/44144 |
| sulfur | 10081.0 | 6.0 | https://www.openml.org/d/23515 | https://www.openml.org/d/44145 |
| medical_charges | 163065.0 | 5.0 | https://www.openml.org/d/42720 | https://www.openml.org/d/44146 |
| MiamiHousing2016 | 13932.0 | 14.0 | https://www.openml.org/d/43093 | https://www.openml.org/d/44147 |
| superconduct | 21263.0 | 79.0 | https://www.openml.org/d/43174 | https://www.openml.org/d/44148 |

### A.1.3  Categorical classification

OpenML benchmark: `https://www.openml.org/search?type=benchmark&sort=date&study_type=task&id=334`

| dataset_name | n_samples | n_features | original_link | new_link |
|---|---|---|---|---|
| electricity | 38474.0 | 8.0 | https://www.openml.org/d/151 | https://www.openml.org/d/44156 |
| eye_movements | 7608.0 | 23.0 | https://www.openml.org/d/1044 | https://www.openml.org/d/44157 |
| covertype | 423680.0 | 54.0 | https://www.openml.org/d/1596 | https://www.openml.org/d/44159 |
| albert | 58252.0 | 31.0 | https://www.openml.org/d/41147 | https://www.openml.org/d/45035 |
| compas-two-years | 4966.0 | 11.0 | https://www.openml.org/d/42192 | https://www.openml.org/d/45039 |
| default-of-credit-card-clients | 13272.0 | 21.0 | https://www.openml.org/d/42477 | https://www.openml.org/d/45036 |
| road-safety | 111762.0 | 32.0 | https://www.openml.org/d/42803 | https://www.openml.org/d/45038 |

### A.1.4 Categorical regression

OpenML benchmark: `https://www.openml.org/search?type=benchmark&study_type=task&sort=tasks_included&id=335`

| dataset_name | n_samples | n_features | original_link | new_link |
|---|---|---|---|---|
| topo_2_1 | 8885.0 | 255.0 | https://www.openml.org/d/422 | https://www.openml.org/d/45041 |
| analcatdata_supreme | 4052.0 | 7.0 | https://www.openml.org/d/504 | https://www.openml.org/d/44055 |
| visualizing_soil | 8641.0 | 4.0 | https://www.openml.org/d/688 | https://www.openml.org/d/44056 |
| delays_zurich_transport | 5465575.0 | 12.0 | https://www.openml.org/d/40753 | https://www.openml.org/d/45045 |
| diamonds | 53940.0 | 9.0 | https://www.openml.org/d/42225 | https://www.openml.org/d/44059 |
| Allstate_Claims_Severity | 188318.0 | 124.0 | https://www.openml.org/d/42571 | https://www.openml.org/d/45046 |
| Mercedes_Benz_Greener_Manufacturing | 4209.0 | 359.0 | https://www.openml.org/d/42570 | https://www.openml.org/d/44061 |
| Brazilian_houses | 10692.0 | 11.0 | https://www.openml.org/d/42688 | https://www.openml.org/d/44062 |
| Bike_Sharing_Demand | 17379.0 | 11.0 | https://www.openml.org/d/42712 | https://www.openml.org/d/44063 |
| Airlines_DepDelay_1M | 1000000.0 | 5.0 | https://www.openml.org/d/42721 | https://www.openml.org/d/45047 |
| nyc-taxi-green-dec-2016 | 581835.0 | 16.0 | https://www.openml.org/d/42729 | https://www.openml.org/d/44065 |
| abalone | 4177.0 | 8.0 | https://www.openml.org/d/42726 | https://www.openml.org/d/45042 |
| house_sales | 21613.0 | 17.0 | https://www.openml.org/d/42731 | https://www.openml.org/d/44066 |
| seattlecrime6 | 52031.0 | 4.0 | https://www.openml.org/d/42496 | https://www.openml.org/d/45043 |
| medical_charges | 163065.0 | 5.0 | https://www.openml.org/d/42720 | https://www.openml.org/d/45048 |
| particulate-matter-ukair-2017 | 394299.0 | 6.0 | https://www.openml.org/d/42207 | https://www.openml.org/d/44068 |
| SGEMM_GPU_kernel_performance | 241600.0 | 9.0 | https://www.openml.org/d/43144 | https://www.openml.org/d/44069 |

## A.2 More benchmarks

### A.2.1 Results as a function of random search time

In Figure 6 and Figure 7, we present the same results that in section 4.2, but as a function of random search *time* instead of random search *iterations*.

**Details** Evaluation and training time are added. Time is averaged among folds, and cumulative time spent on random search is binned into 20 bins. Deep learning models are run on GPUs, and tree-based models on CPUs (see A.3. Note that we train tree-based models on single CPUs, which means we overestimate the time taken by an algorithm like random forest compared to what practitioners would typically experience). We present this comparison to give a rough sense of the speed difference between tree-based models and neural networks, but this should not be considered a rigorous comparison of the speed of different models, as we use different types of GPUs and CPUs.

**Results** Looking at the results as a function of random search time rather than random search iterations makes tree-based models superiority even more striking. Neural networks and tree-based models were close for some benchmarks after a small number of iterations, but for the same amount of time spent on random search, tree-based models scores are always high above neural networks.

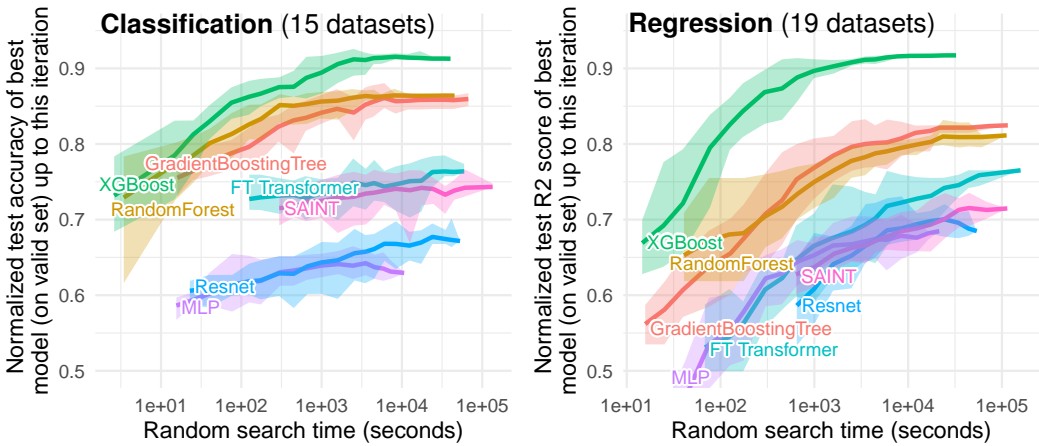

Figure 6: **Time benchmark on medium-sized datasets, with only numerical features**. The first random search iteration corresponds to default hyperparameters. Each value corresponds to the test score of the best model (on the validation set) after a specific time spent doing random search, averaged on 15 shuffles of the random search order. The ribbon corresponds to the minimum and maximum scores on these 15 shuffles.

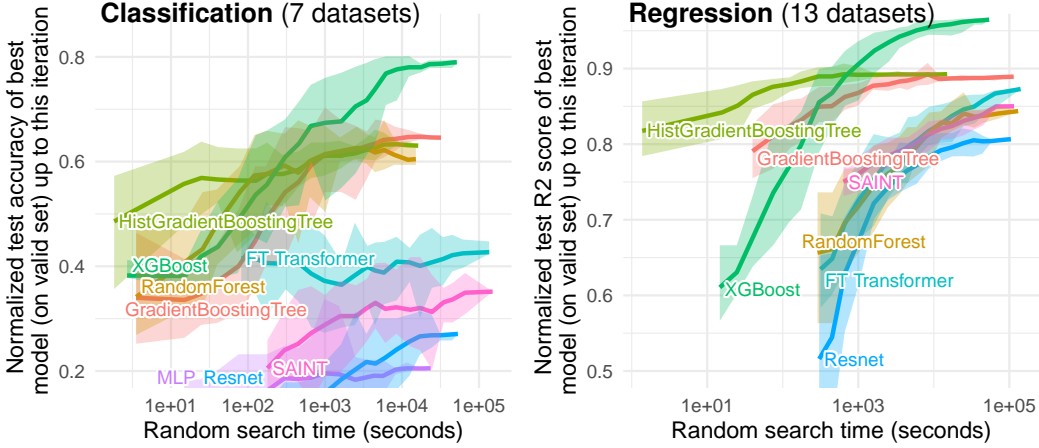

Figure 7: **Time benchmark on medium-sized datasets, with both numerical and categorical features**. The first random search iteration corresponds to default hyperparameters. Each value corresponds to the test score of the best model (on the validation set) after a specific time spent doing random search, averaged on 15 shuffles of the random search order. The ribbon corresponds to the minimum and maximum scores on these 15 shuffles.

### A.2.2 Large-sized datasets

We extend our benchmark to large-scale datasets: in Figures 8, 9, 10 and 11, we compare the results of our models on the same set of datasets, in large-size (train set truncated to 50,000 samples) and medium-size (train set truncated to 10,000 samples) settings.

We only keep datasets with more than 50,000 samples and restrict the train set size to 50,000 samples (vs 10,000 samples for the medium-sized benchmark). Unfortunately, this excludes a lot of datasets, which makes the comparison less clear. However, it seems that, in some cases, increasing the train set size reduces the gap between neural networks and tree-based models. We leave a rigorous study of this trend to future work. In A.5.2, we study how many datasets each step of the datasets filtering process remove, and find that no single step is responsible for the difficulty of gathering a large number of large-scaled datasets.

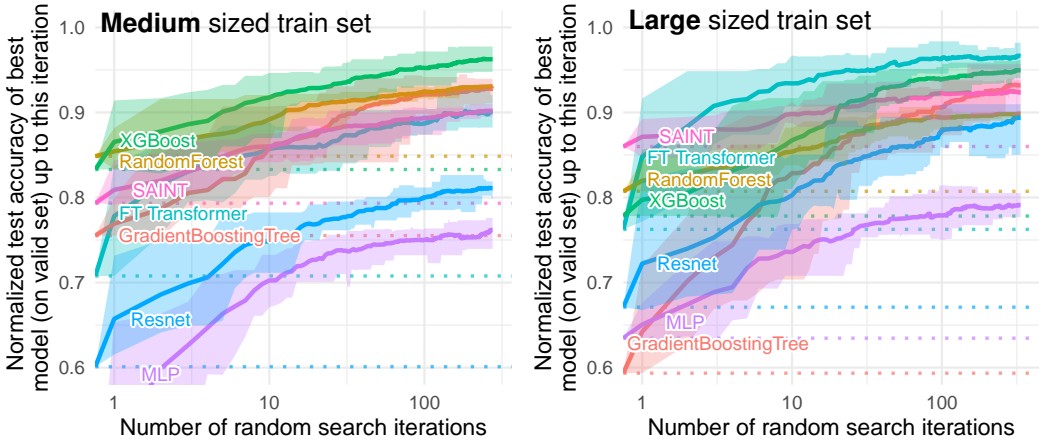

Figure 8: **Comparison of accuracies on 4 classification tasks for different train set sizes, with only numerical features**. Only datasets with more than 50,000 samples were kept, and the train set size was truncated to either 10,000 samples or 50,000 samples. Dotted lines correspond to the score of the default hyperparameters, which is also the first random search iteration. Each value corresponds to the test score of the best model (on the validation set) after a specific number of random search iterations, averaged on 15 shuffles of the random search order. The ribbon corresponds to the minimum and maximum scores on these 15 shuffles.

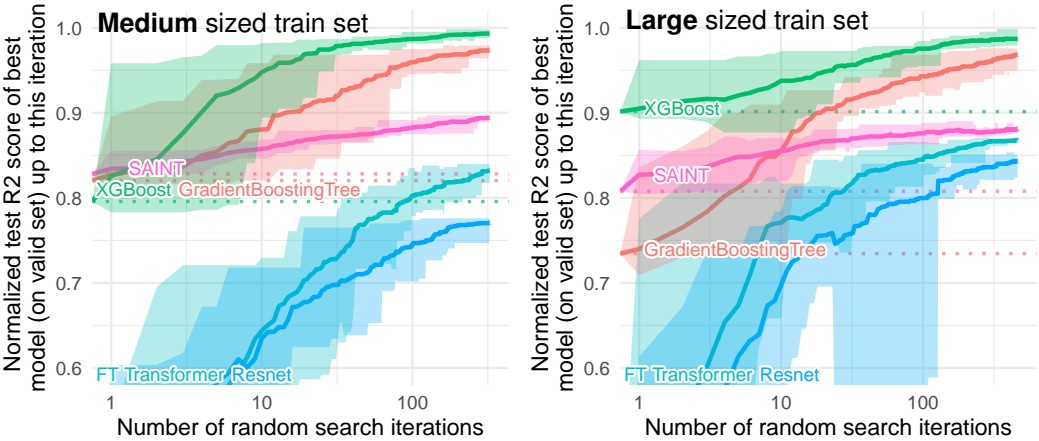

Figure 9: **Comparison of R2 scores on 3 regression tasks for different train set sizes, with only numerical features**. Only datasets with more than 50,000 samples were kept, and the train set size was truncated to either 10,000 samples or 50,000 samples. Dotted lines correspond to the score of the default hyperparameters, which is also the first random search iteration. Each value corresponds to the test score of the best model (on the validation set) after a specific number of random search iterations, averaged on 15 shuffles of the random search order. The ribbon corresponds to the minimum and maximum scores on these 15 shuffles.

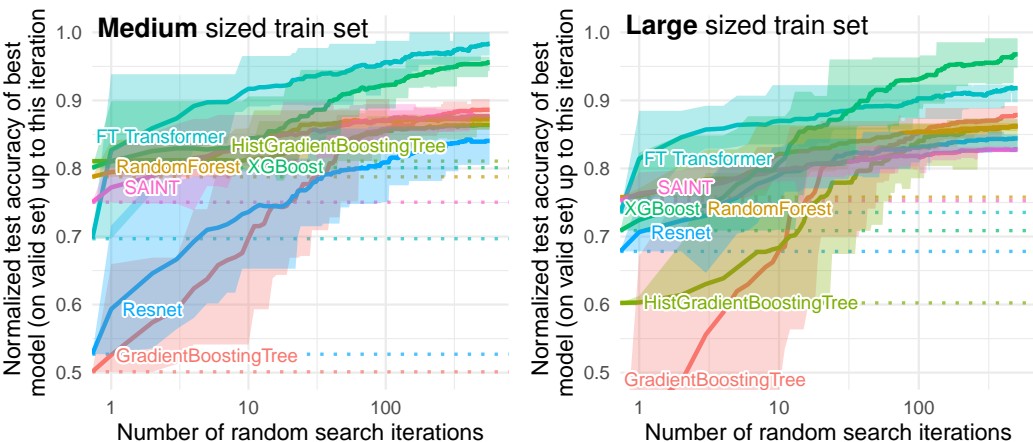

Figure 10: **Comparison of accuracies on 2 classification tasks for different train set sizes, with both numerical and categorical features**. Only datasets with more than 50,000 samples were kept, and the train set size was truncated to either 10,000 samples or 50,000 samples. Dotted lines correspond to the score of the default hyperparameters, which is also the first random search iteration. Each value corresponds to the test score of the best model (on the validation set) after a specific number of random search iterations, averaged on 15 shuffles of the random search order. The ribbon corresponds to the minimum and maximum scores on these 15 shuffles.

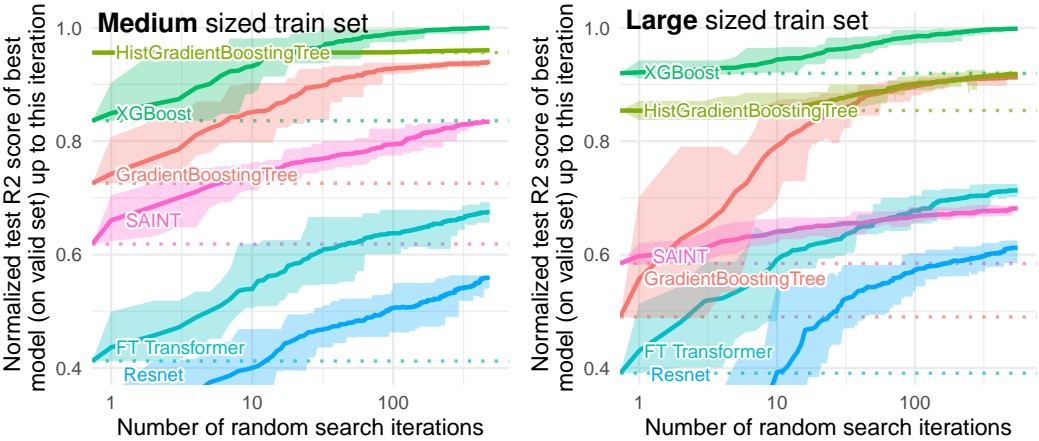

Figure 11: **Comparison of R2 scores on 5 regression tasks for different train set sizes, with both numerical and categorical features**. Only datasets with more than 50,000 samples were kept, and the train set size was truncated to either 10,000 samples or 50,000 samples. Dotted lines correspond to the score of the default hyperparameters, which is also the first random search iteration. Each value corresponds to the test score of the best model (on the validation set) after a specific number of random search iterations, averaged on 15 shuffles of the random search order. The ribbon corresponds to the minimum and maximum scores on these 15 shuffles.

## A.3 More details on benchmark

**Train / Validation / Test split**    We take 70% of samples for the train set (or the percentage which corresponds to the maximum train set size if 70% is too high). Of the remaining 30%, we take 30% for the validation set, and 70% for the test set. The validation and test sets are truncated to 50,000 samples for speed. Note that the validation set is only used to select the best performing hyperparameter combination during the random search, and is distinct from the validation set used for early stopping (which is part of the train set).

**Number of folds**    For each dataset and hyperparameters combination, we vary the number of folds used for our algorithms evaluation depending on the number of testing samples:

- If We have more than 6000 samples, we evaluate our algorithms on 1 fold.
- If we have between 3000 and 6000 samples, we evaluate our algorithms on 2 folds.
- If we have between 1000 and 3000 samples, we evaluate our algorithms on 3 folds.
- If we have less than 1000 testing samples, we evaluate our algorithms on 5 folds.

Every algorithm and hyperparameters combination is evaluated on the same folds.

**Hardware**    For all our benchmarks and experiments, we use the hardware below. The hardware was chosen based on availability, with Neural Networks always running on GPU and tree-based models running on CPU.

GPUs: NVIDIA Quadro RTX 6000, NVIDIA TITAN Xp, NVIDIA A100, NVIDIA V100, NVIDIA Tesla T4, NVIDIA A40, NVIDIA TITAN RTX, NVIDIA TITAN V

CPUs: AMD EPYC 7742 64-Core Processor, AMD EPYC 7702 64-Core Processor, Intel(R) Xeon(R) CPU E5-2660 v2, Intel(R) Xeon(R) Gold 6226R CPU

**Hyperparameters space**    Hyperparameters spaces are based on Hyperopt-sklearn [Komer et al., 2014] when available, from Gorishniy et al. [2021] and from Borisov et al. [2021]. We made some changes when combining sources, or when the original distribution was not compatible with Weight and Biases sweeps. Furthermore, contrary to Hyperopt-sklearn, we do not tune the number of estimators for tree-based models, but set it to a constant high value (250 for random forest, following Oshiro et al. [2012], 1000 for gradient boosting models) and use early stopping for gradient boosting trees models. We study the impact of this change in A.9.2. The maximum numbers of estimators we choose are small and quite often reached: this is done on purpose, to compare fast tree-based models to slower neural networks. In A.9.1, we study the performance gain we obtain when running our tree-based models for longer.

Default parameters for tree-based models are ScikitLearn's defaults. All neural networks are run for 300 epochs, with early stopping and checkpointing (the best model on the validation set is kept). Early stopping patience is 40 for MLP, Resnet, and FT Transformer, 20 for GradientBoostingTrees and XGBoost, and 10 for SAINT. For model using early stopping, 20% of the training dataset is used as validation set.

| Parameter | Distribution | Default |
|---|---|---|
| Num layers | UniformInt $[1, 6]$ | 3 |
| Feature embedding size | UniformInt $[64, 512]$ | 192 |
| Residual dropout Uniform | $[0, 0.5]$ | 0 |
| Attention dropout | Uniform $[0, 0.5]$ | 0.2 |
| FFN dropout | Uniform $[0, 0.5]$ | 0.1 |
| FFN factor | Uniform $[2/3, 8/3]$ | 4/3 |
| Learning rate | LogUniform$[1e-5, 1e-3]$ | $1e-4$ |
| Weight decay | LogUniform $[1e-6, 1e-3]$ | $1e-5$ |
| kv compression | [True, False] | True |
| kv compression sharing | [headwise, key-value] | headwise |
| Learning rate scheduler | [True, False] | False |
| Batch size | $[256, 512, 1024]$ | 512 |

Table 1: FT Transformer hyperparameters space

| Parameter | Distribution | Default |
|---|---|---|
| Num layers | UniformInt $[1, 16]$ | 8 |
| Layer size | UniformInt $[64, 1024]$ | 256 |
| Hidden factor | Uniform $[1, 4]$ | 2 |
| Hidden dropout | $[0, 0.5]$ | 0.2 |
| Residual dropout | Uniform$[0, 0.5]$ | 0.2 |
| Learning rate | LogUniform$[1e-5, 1e-2]$ | $1e-3$ |
| Weight decay | LogUniform $[1e-8, 1e-3]$ | $1e-7$ |
| Category embedding size | UniformInt $[64, 512]$ | 128 |
| Normalization | [batchnorm, layernorm] | batchnorm |
| Learning rate scheduler | [True, False] | True |
| Batch size | $[256, 512, 1024]$ | 512 |

Table 2: Resnet hyperparameters space

| Parameter | Distribution | Default |
|---|---|---|
| Num layers | UniformInt $[1, 8]$ | 4 |
| Layer size | UniformInt $[16, 1024]$ | 256 |
| Dropout | $[0, 0.5]$ | 0.2 |
| Learning rate | LogUniform$[1e-5, 1e-2]$ | $1e-3$ |
| Category embedding size | UniformInt $[64, 512]$ | 128 |
| Learning rate scheduler | [True, False] | True |
| Batch size | $[256, 512, 1024]$ | 512 |

Table 3: MLP hyperparameters space

| Parameter | Distribution | Default |
|---|---|---|
| Num layers | UniformInt $[1, 2, 3, 6, 12]$ | 3 |
| Num heads | $[2, 4, 8]$ | 4 |
| Layer size | UniformInt $[32, 64, 128]$ | 128 |
| Dropout | $[0, 0.1, 0.2, 0.3, 0.4, 0.5, 0.6, 0.7, 0.8]$ | 0.1 |
| Learning rate | LogUniform$[1e-5, 1e-3]$ | $3e-5$ |
| Batch size | $[128, 256]$ | 512 |

Table 4: SAINT hyperparameters space

| Parameter | Distribution |
|---|---|
| Max depth | UniformInt[1,11] |
| Num estimators | 1000 |
| Min child weight | LogUniformInt[1, 1e2] |
| Subsample | Uniform[0.5,1] |
| Learning rate | LogUniform[1e-5,0.7] |
| Col sample by level | Uniform[0.5,1] |
| Col sample by tree | Uniform[0.5, 1] |
| Gamma | LogUniform[1e-8,7] |
| Lambda | LogUniform[1,4] |
| Alpha | LogUniform[1e-8,1e2] |

Table 5: XGBoost hyperparameters space

| Parameter | Distribution |
|---|---|
| Max depth | [None, 2, 3, 4] ([0.7, 0.1, 0.1, 0.1]) |
| Num estimators | 250 |
| Criterion | [gini, entropy] (classif) [squared_error, absolute_error] (regression) |
| Max features | [sqrt, sqrt, log2, None, 0.1, 0.2, 0.3, 0.4, 0.5, 0.6, 0.7, 0.8, 0.9] |
| Min samples split | [2, 3] ([0.95, 0.05]) |
| Min samples leaf | LogUniformInt[1.5, 50.5] |
| Boostrap | [True, False] |
| Min impurity decrease | [0.0, 0.01, 0.02, 0.05] ([0.85, 0.05, 0.05, 0.05]) |

Table 6: RandomForest hyperparameters space

| Parameter | Distribution |
|---|---|
| Loss | [deviance, exponential] (classif) [squared_error, absolute_error, huber] (regression) |
| Learning rate | LogNormal[log(0.01), log(10)] |
| Subsample | Uniform[0.5, 1] |
| Num estimators | 1000 |
| Criterion | [friedman_mse, squared_error] |
| Max depth | [None, 2, 3, 4, 5] ([0.1, 0.1, 0.6, 0.1, 0.1]) |
| Min samples split | [2, 3] ([0.95, 0.05]) |
| Min samples leaf | LogUniformInt[1.5, 50.5] |
| Min impurity decrease | [0.0, 0.01, 0.02, 0.05] ([0.85, 0.05, 0.05, 0.05]) |
| Max leaf nodes | [None, 5, 10, 15] ([0.85, 0.05, 0.05, 0.05]) |

Table 7: GradientBoosting hyperparameters space

| Parameter | Distribution |
|---|---|
| Loss | [squared_error, absolute_error] (regression) |
| Learning rate | LogNormal[log(0.01), log(10)] |
| Max iter | 1000 |
| Max depth | [None, 2, 3, 4] ([0.1, 0.1, 0.7, 0.1]) |
| Min samples leaf | NormalInt[20, 2] |
| Max leaf nodes | NormalInt[31, 5] |

Table 8: HistGradientBoosting hyperparameters space

**Other details** To run the random searches, we use the "sweep" functionality of Weight and Biases [Biewald, 2020].

**Dataset by dataset** We show all unnormalized benchmark results dataset by dataset: Figure 12, 13, 14 and 15 for the medium-size setting, and Figure 16 and 17 for the large-size setting.

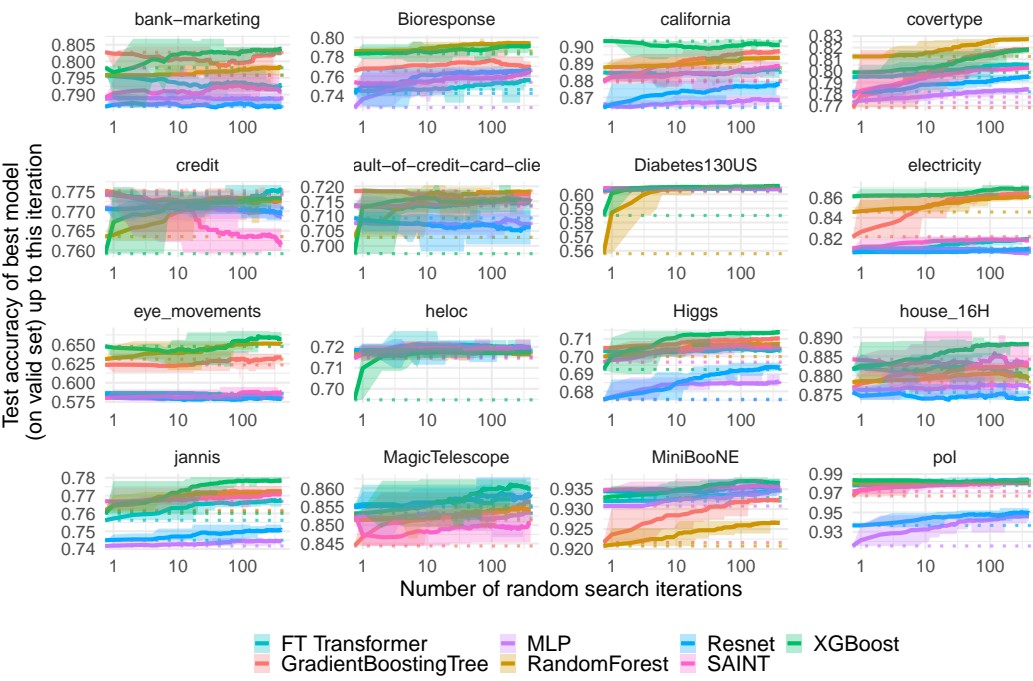

Figure 12: **Unnormalized benchmark results for classification tasks on numerical features only.** Medium-sized setting.

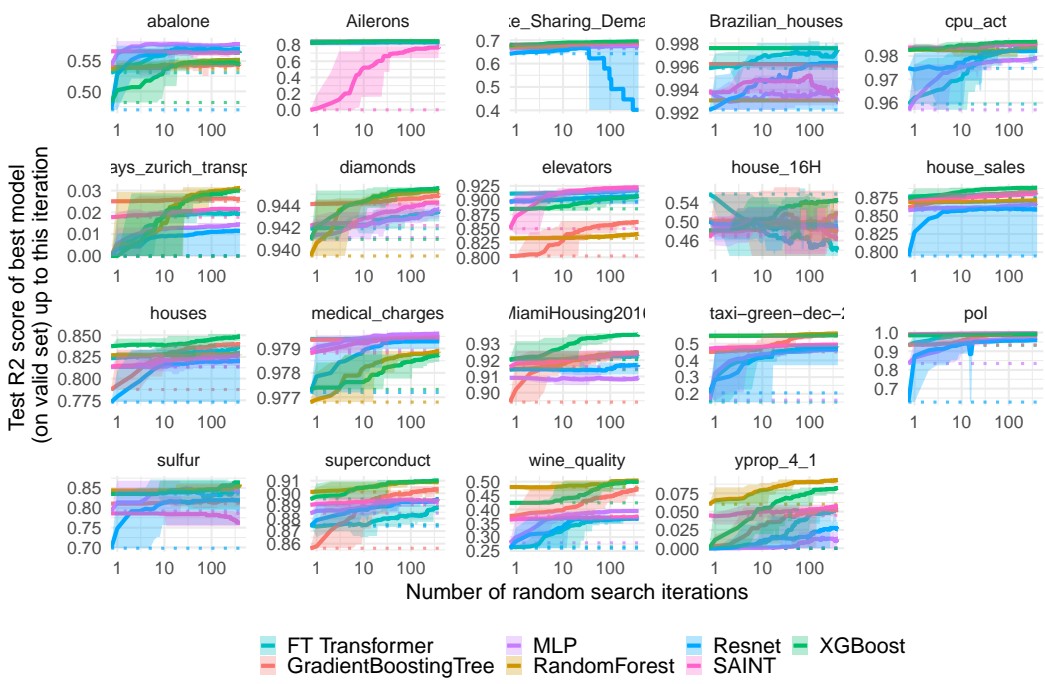

Figure 13: **Unormalized benchmark results for regression tasks on numerical features only.** Medium-sized setting. Negative values are truncated to zero to make the plots easier to read.

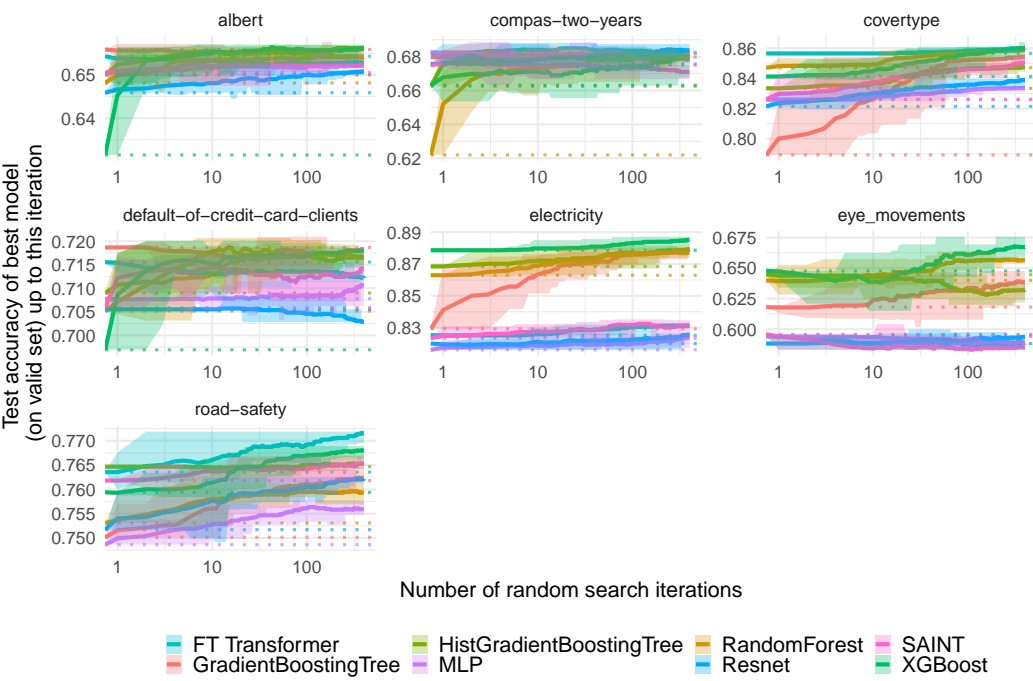

Figure 14: **Unormalized benchmark results for classification tasks on both categorical and numerical features.** Medium-sized setting.

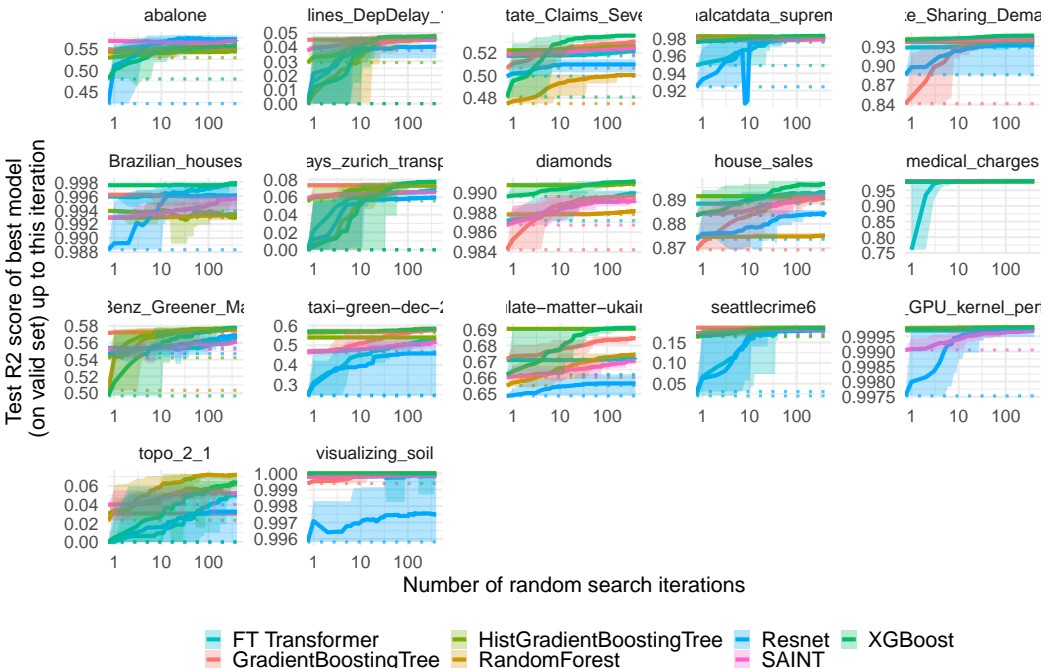

Figure 15: **Unormalized benchmark results for regression tasks on both categorical and numerical features.** Medium-sized setting. Negative values are truncated to zero to make the plots easier to read.

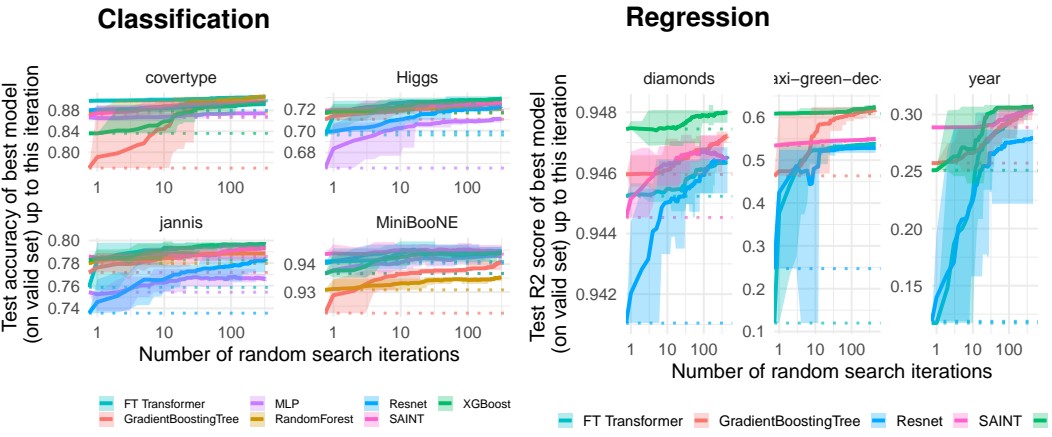

Figure 16: **Unormalized benchmark results for large scale datasets, for numerical features only.**

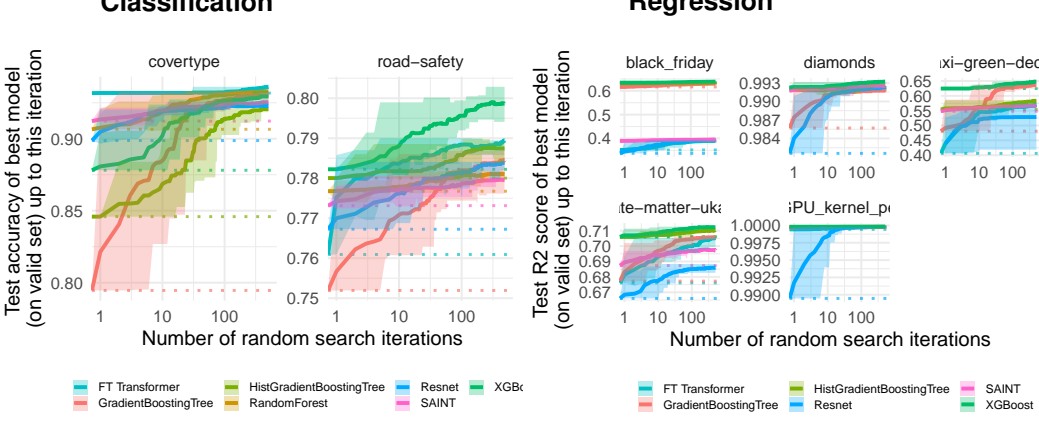

Figure 17: **Unormalized benchmark results for large-scale datasets, for both numerical and categorical features.**

### A.4 More details on experiments

In this section, we give more details on the choices we made when creating our experiments. As results may sometimes be easier to interpret before aggregation across datasets, we also show the results of each experiment on each dataset.

#### A.4.1 Finding 1: Neural networks are biased to overly smooth solutions

**Smoothing** We use a Gaussian smoothing Kernel

$$K(\mathbf{x}^*, \mathbf{x}) = \exp\left(-\frac{1}{2}(\mathbf{x}^* - \mathbf{x})^{\mathrm{T}}\boldsymbol{\Sigma}^{-1}(\mathbf{x}^* - \mathbf{x})\right)$$

with $\boldsymbol{\Sigma}$ the estimated empirical covariance multiplied by the "squared lengthscale". The transformed target on the train set becomes:

$$\tilde{Y}(X_i) = \frac{\sum_{j=1}^{N} K(X_i, X_j) Y(X_j)}{\sum_{j=1}^{N} K(X_i, X_j)}$$

with $(X_1, ..X_N)$ the training set covariates and $(Y_1, ..., Y_N)$ the training set original targets.

**More details**

- We restrict all datasets to their 5 most important features (according to a RandomForest feature importance ranking). This makes the smoothing easier, as kernel smoothing can be hard in high-dimension, while keeping enough features to produce interesting results.
- We estimate the covariance matrix of these features through ScikitLearn's `MinCovDet`, which is more robust to outliers than the empirical covariance.

Raw results are shown in Figure 18 dataset by dataset.

**Examples of irregular patterns** Figure 19 shows the decision boundaries of a default MLP and a default RandomForest on the 2 most important features of the *electricity* dataset. The RandomForest achieve a perfect training accuracy and a test accuracy (85%) higher than the MLP (80%). The features are Gaussienied and we show a zoomed-in part of the feature space. In this part, we can see that the RandomForest is able to learn irregular patterns on the x axis (which corresponds to the *date* feature) that the MLP does not learn. We show this difference for default hyperparameters but it seems to us that this is a typical behavior of neural networks, and it is actually hard, albeit not impossible, to find hyperparameters to successfully learn these patterns.

**Quantitative analysis**

We run the following linear regression in R, using the same data than those used in the corresponding plot (using the normalized test score of best model (on the validation set) after 60 random search iterations).

```
lm(mean_test_score~dataset + model_name + lengthscale +
model_name * lengthscale)}
```

We show the results rounded to the third decimal, with the coefficients for each dataset not displayed for lisibility. The results match our plots: increasing the lengthscale decreases performances (statistically) significantly more for tree based models, (the interaction term between the `modelname` variable and the `lengthscale` variable is significantly more negative for tree-based models).

| term | estimate | std.error | statistic | p.value |
|---|---|---|---|---|
| (Intercept) | 0.551 | 0.009 | 58.558 | 0 |
| model_nameGradientBoostingTree | 0.073 | 0.008 | 9.082 | 0 |
| model_nameRandomForest | 0.028 | 0.008 | 3.537 | 0 |
| model_nameResnet | -0.04 | 0.008 | -5.044 | 0 |
| lengthscale | -0.411 | 0.022 | -18.492 | 0 |
| model_nameGradientBoostingTree:lengthscale | -0.3 | 0.031 | -9.557 | 0 |
| model_nameRandomForest:lengthscale | -0.197 | 0.031 | -6.278 | 0 |
| model_nameResnet:lengthscale | 0.024 | 0.031 | 0.772 | 0.44 |

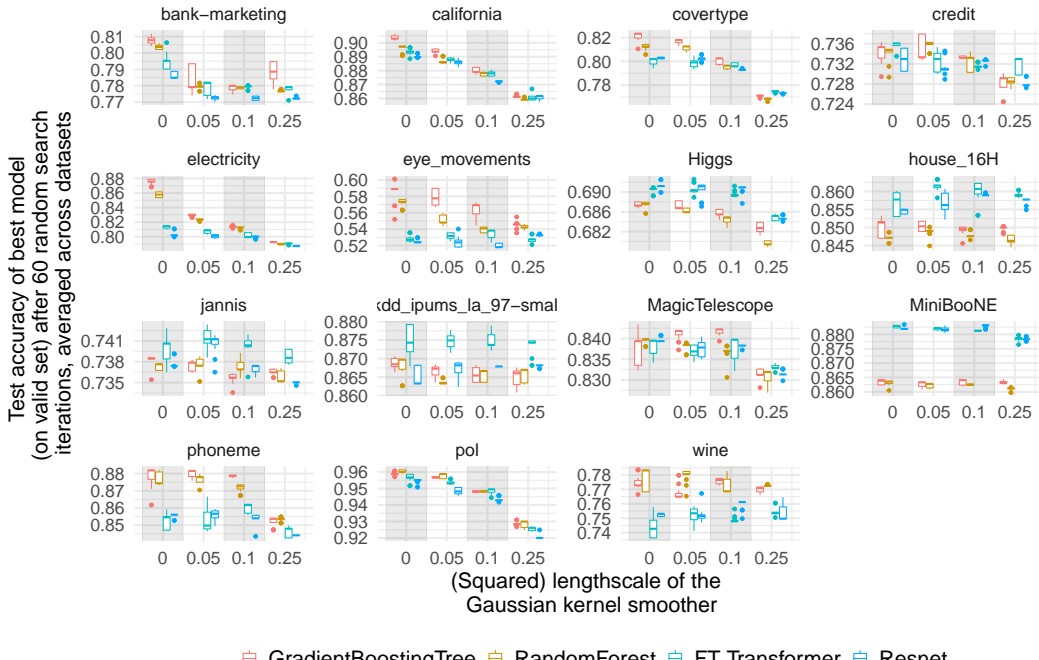

Figure 18: **Test accuracy of different models for varying smoothing of the target function on the train set**. We smooth the target function through a Gaussian Kernel smoother, whose covariance matrix is the data covariance, multiplied by the (squared) lengthscale of the Gaussian kernel smoother. A lengthscale of 0 corresponds to no smoothing (the original data). All features have been Gaussienized before the smoothing through ScikitLearn's QuantileTransformer. The boxplots represent the distribution of accuracies across 15 re-orderings of the random search. Same experiment than Fig. 2, shown for each dataset without score normalization

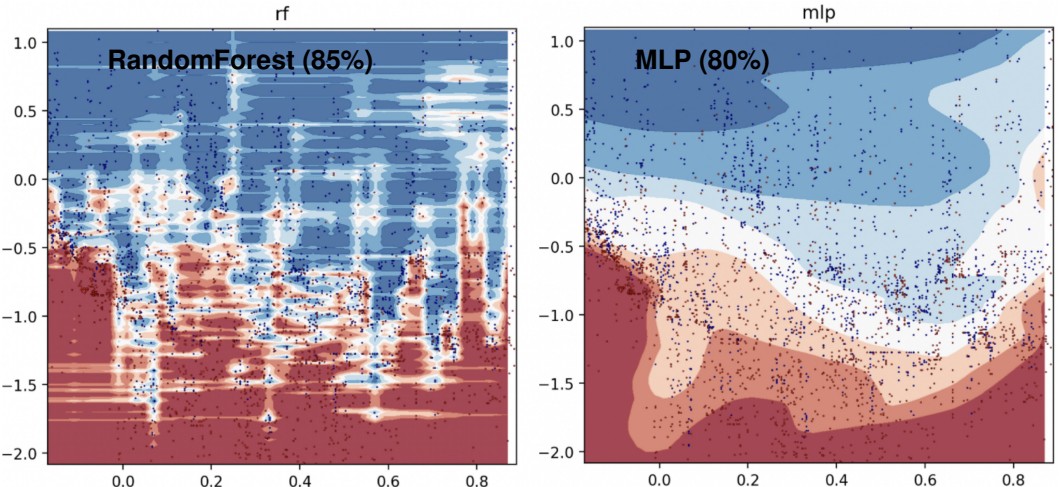

Figure 19: Decision boundaries of a default MLP and RandomForest for the 2 most important features of the *electricity* dataset

## A.4.2 Finding 2: Uninformative features affect more MLP-like neural networks

**Tabular datasets contain a lot of uninformative features**  Raw results are shown in Figure 20 dataset by dataset.

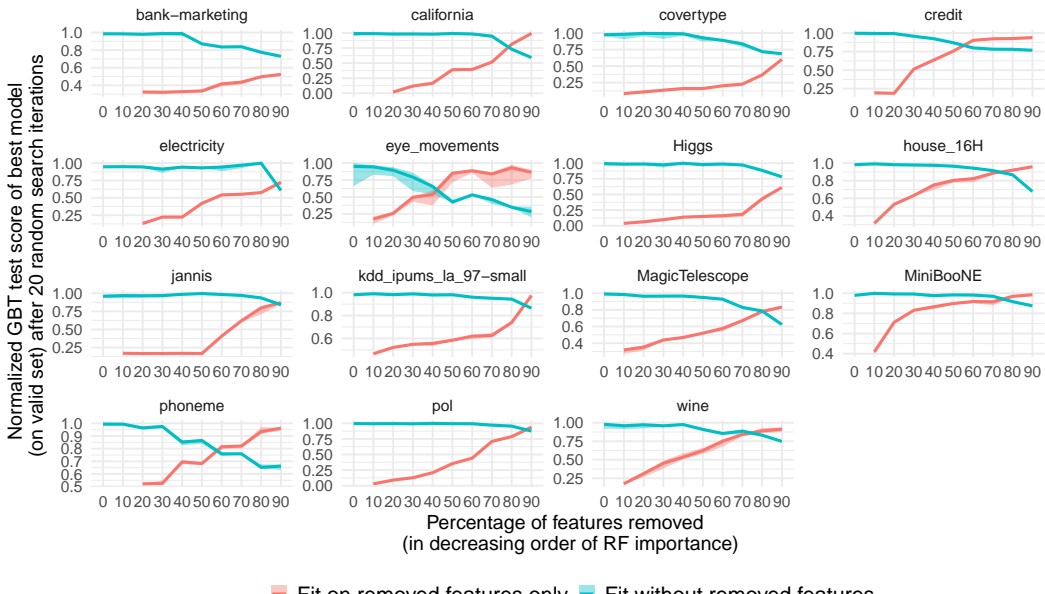

Figure 20: **Test accuracy of a GBT for varying proportions of removed features**. Features are removed in increasing order of feature importance (computed with a Random Forest), and the two lines correspond to the accuracy using the (most important) kept features (blue) or the (least important) removed features (red). Scores are normalized between 0 (random chance) and 1 (best score among all hyperparameters). These scores are averaged across 30 random search orders, and the ribbons correspond to the minimum and maximum values among these 30 orders. Same experiment than Fig. 3, shown for each dataset. Note that axes do not always start at zero.

**Uninformative features affect more MLP-like neural networks** Raw results are shown in Figure 21 (uninformative features added) and Figure 22 (uninformative features removed).

**Quantitative analysis**

We run the following linear regressions in R, using the same data than those used in the corresponding plot (using the normalized test score of best model (on the validation set) after 20 random search iterations).

```
lm(mean_test_score~dataset + model_name + prop_removed +
model_name * prop_removed)
```

and

```
lm(mean_test_score~dataset + model_name + prop_added +
model_name * prop_added)
```

We show the results rounded to the third decimal, with the coefficients for each dataset not displayed for lisibility.

The results match our plots:

- The decrease in performance when removing uninformative features is (statistically) significantly more negative for tree-based models (the interaction term `model_name * prop_remove` is (statistically) significantly negative).

- The decrease in performance when adding uniformative features is (statistically) significantly more negative for Resnets than for FT Transformer, whereas tree-based models' decrease is not (statistically) significantly different.

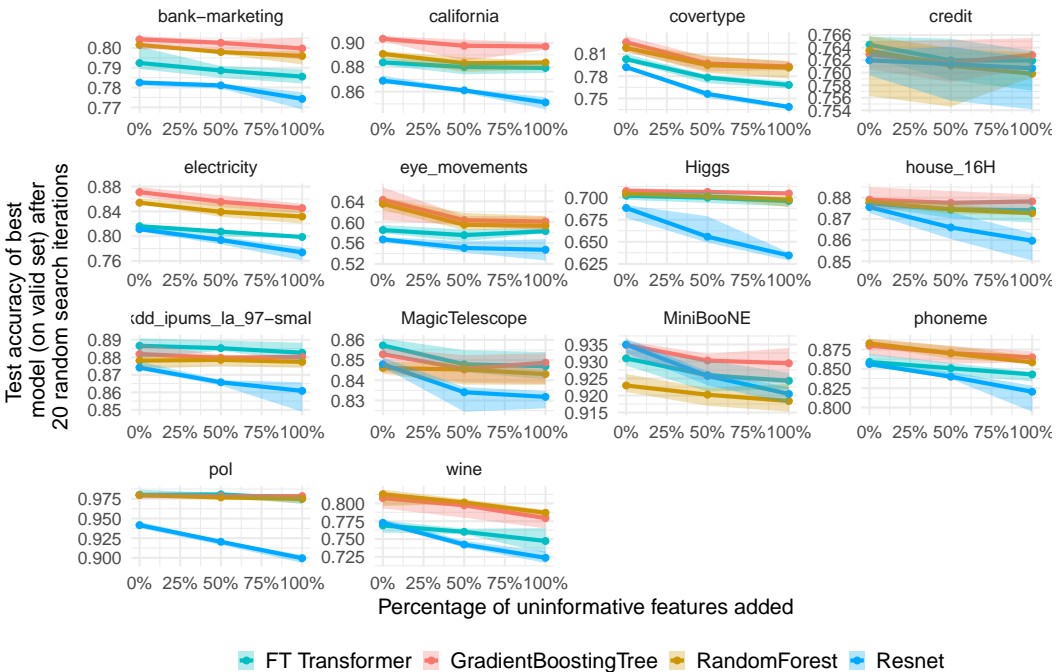

Figure 21: **Test accuracy changes when adding uninformative features**. Added features are sampled from standard Gaussians uncorrelated with the target and with other features. Ribbons correspond to the minimum and maximum score among the 30 different random search reorders (starting with the default models). Same experiment that in Figure 4 (b), shown for each dataset without score normalization.

| term | estimate | std.error | statistic | p.value |
|------|----------|-----------|-----------|---------|
| (Intercept) | 0.824 | 0.007 | 115.073 | 0 |
| model_nameGradientBoostingTree | 0.142 | 0.007 | 19.535 | 0 |
| model_nameRandomForest | 0.089 | 0.007 | 12.27 | 0 |
| model_nameResnet | -0.094 | 0.007 | -12.991 | 0 |
| prop_removed | -0.681 | 0.01 | -70.716 | 0 |
| model_nameGradientBoostingTree:prop_removed | -0.101 | 0.014 | -7.426 | 0 |
| model_nameRandomForest:prop_removed | -0.047 | 0.014 | -3.474 | 0.001 |
| model_nameResnet:prop_removed | 0.113 | 0.014 | 8.327 | 0 |

| term | estimate | std.error | statistic | p.value |
|------|----------|-----------|-----------|---------|
| (Intercept) | 0.951 | 0.018 | 53.335 | 0 |
| model_nameGradientBoostingTree | 0.133 | 0.023 | 5.831 | 0 |
| model_nameRandomForest | 0.098 | 0.023 | 4.317 | 0 |
| model_nameResnet | 0.06 | 0.023 | 2.639 | 0.008 |
| prop_added | -0.101 | 0.01 | -9.737 | 0 |
| model_nameGradientBoostingTree:prop_added | -0.007 | 0.015 | -0.49 | 0.624 |
| model_nameRandomForest:prop_added | -0.014 | 0.015 | -0.967 | 0.334 |
| model_nameResnet:prop_added | -0.156 | 0.015 | -10.641 | 0 |

### A.4.3 Finding 3: Data are non invariant by rotation, so should be learning procedures

**Details** Random rotation were computed using Scipy's [Virtanen et al., 2020] `stats.special_ortho_group.rvs`.

Raw results are shown in Figure 23 (with all features) and Figure 24 (with 50% features) dataset by dataset.

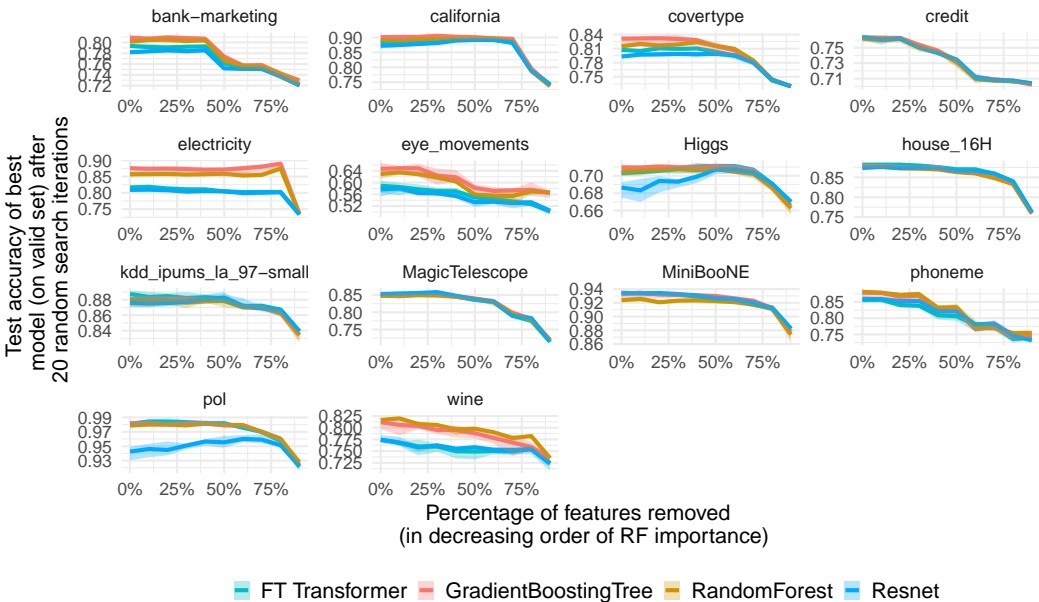

Figure 22: **Test accuracy changes when removing uninformative features**. Features are removed in increasing order of feature importance (computed with a Random Forest). Ribbons correspond to the minimum and maximum score among the 30 different random search reorders (starting with the default models). Same experiment that in Figure 4 (a), shown for each dataset without score normalization.

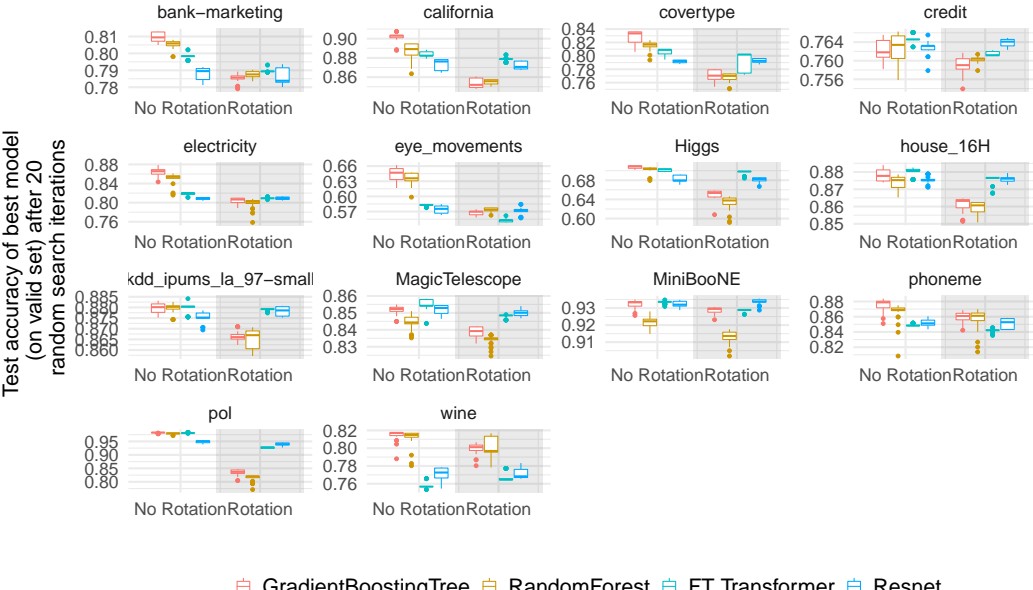

Figure 23: **Test accuracy of different models when randomly rotating our datasets**. All features are Gaussianized before the random rotations. The scores are averaged across datasets, and the boxes depict the distribution across random search shuffles. Same experiment that in 5 (Left), shown for each dataset without score normalization.

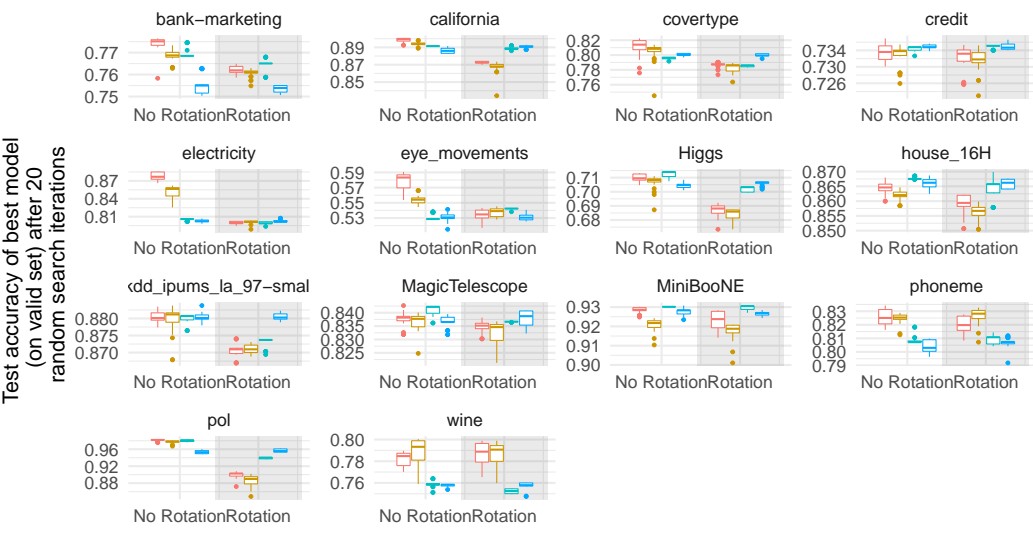

Figure 24: **Test accuracy of different models when randomly rotating our datasets, with 50% features removed**. All features are Gaussianized before the random rotations. The removed features are the least important half (according to a RandomForest), and are removed before the rotation. The scores are averaged across datasets, and the boxes depict the distribution across random search shuffles. Same experiment that in 5 (Right), shown for each dataset without score normalization.

**Quantitative analysis**   We run the following linear regression in R, using the same data than those used in the corresponding plot (using the normalized test score of best model (on the validation set) after 20 random search iterations).

```
lm(mean_test_score~dataset + model_name + rotation + features_removed +
rotation * model_name + rotation * features_removed)
```

, with `features_removed` and `rotation` being indicator variables indicating if 50% of features where removed and if the dataset was randomly rotated.

We show the results rounded to the third decimal, with the coefficients for each dataset not displayed for lisibility.

The results match our plots: rotating the datasets decreases tree-based models' performance more than neural networks', and rotations' impact is attenuated if 50% of features have been removed. All these effects are statistically significant.

| term | estimate | std.error | statistic | p.value |
|---|---|---|---|---|
| (Intercept) | 0.802 | 0.006 | 123.653 | 0 |
| model_nameGradientBoostingTree | 0.129 | 0.005 | 23.83 | 0 |
| model_nameRandomForest | 0.074 | 0.005 | 13.615 | 0 |
| model_nameResnet | -0.046 | 0.005 | -8.429 | 0 |
| rotationTRUE | -0.085 | 0.006 | -14.114 | 0 |
| features_removedTRUE | -0.009 | 0.004 | -2.363 | 0.018 |
| model_nameGradientBoostingTree:rotationTRUE | -0.173 | 0.008 | -22.595 | 0 |
| model_nameRandomForest:rotationTRUE | -0.138 | 0.008 | -18.073 | 0 |
| model_nameResnet:rotationTRUE | 0.062 | 0.008 | 8.117 | 0 |
| rotationTRUE:features_removedTRUE | 0.051 | 0.005 | 9.456 | 0 |

## A.5   Dataset filtering

We show here each step of the dataset selection process, as explained in 3, as Sankey diagrams. This can be useful to understand which step removes the biggest number of datasets from our benchmarks.

### A.5.1   Medium datasets

**Details**   The left side of the Sankey plot starts from the set of datasets composed of all OpenML datasets plus datasets we found on our own, restricted to those with more than 3K samples and 3 features before filtering.

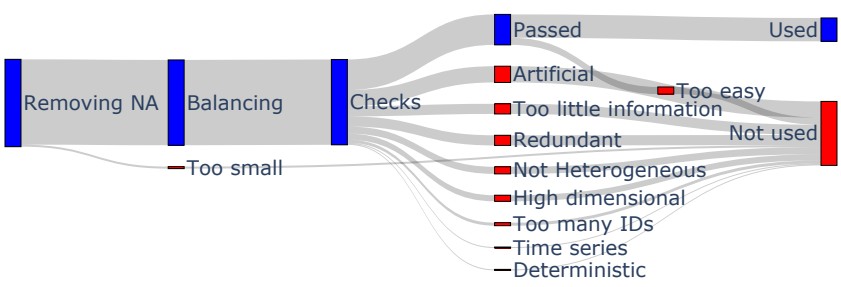

Figure 25: Dataset filtering process for the numerical regression benchmark

### A.5.2   Large datasets

As discussed in A.2.2, we found it hard to gather numerous large-scale (> 50K samples) datasets. The following plots show that this is not due to a single restriction used for our benchmark, but to a

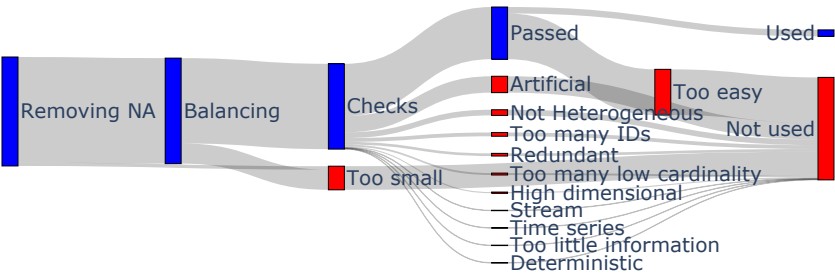

Figure 26: Dataset filtering process for the numerical classification benchmark

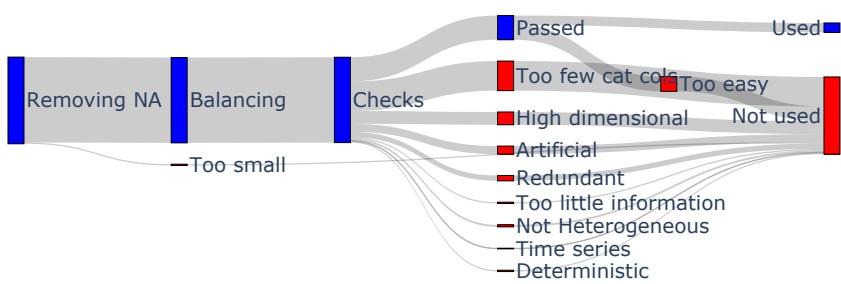

Figure 27: Dataset filtering process for the categorical regression benchmark

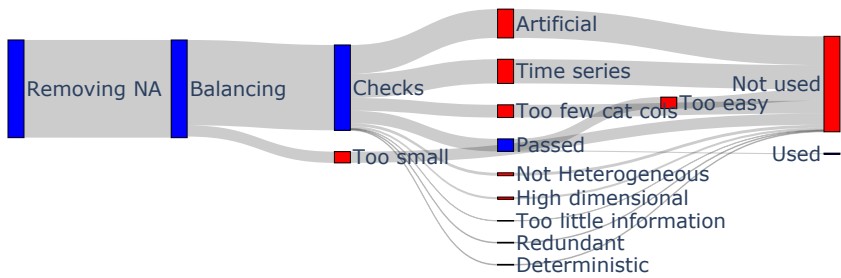

Figure 28: Dataset filtering process for the categorical classification benchmark

conjunction of a lot of different restrictions. This means making benchmarks on large scale datasets would either require finding new datasets we missed, or removing a lot of restrictions at once, which may make the benchmark harder to interpret.

**Details**    The left side of the Sankey plot starts from the set of datasets composed of all OpenML datasets plus datasets we found on our own, restricted to those with more than 50K samples and 3 features before filtering.

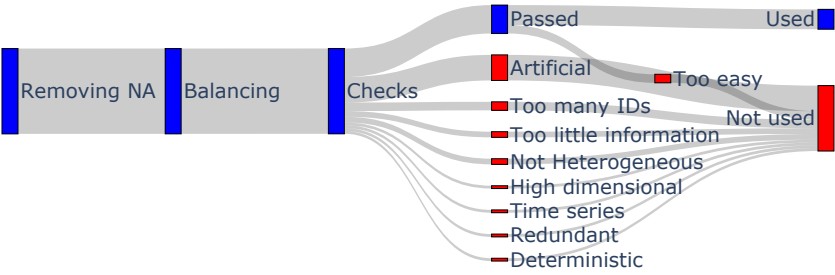

Figure 29: Dataset filtering process for the numerical regression benchmark on large datasets

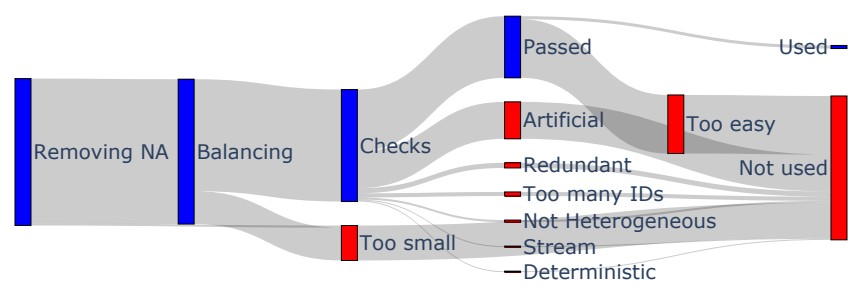

Figure 30: Dataset filtering process for the numerical classification benchmark on large datasets

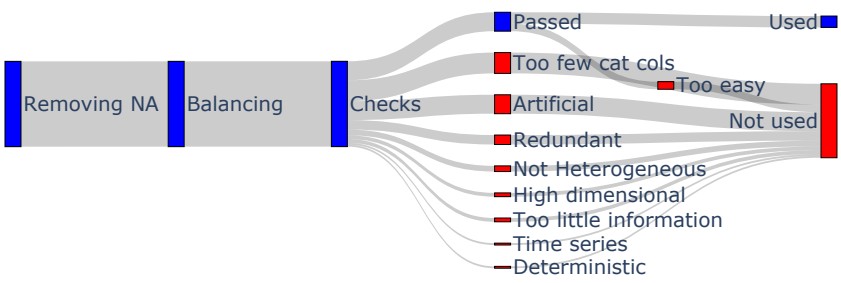

Figure 31: Dataset filtering process for the categorical regression benchmark on large datasets

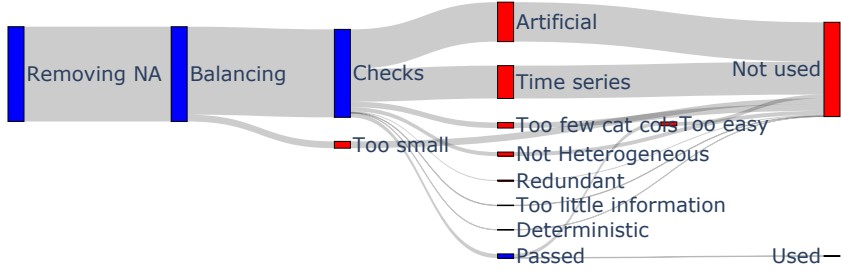

Figure 32: Dataset filtering process for the categorical classification benchmark on large datasets

### A.6 Comparison with other benchmarks

In this section, we study the differences between the set of datasets we selected for our benchmarks with the set selected by previous benchmarks, namely the CC18 benchmark [Bischl et al., 2021] and the AutoML benchmark [Gijsbers et al., 2019]. While the these benchmarks' selection procedures are quite similar to ours, we highlight below a few differences below. Importantly, these small differences in the selection procedure makes for very large difference in the set of datasets selected. As can be seen on the table below, the biggest source of difference between the CC-18 and AutoML benchmarks and ours is that these include many small-scale datasets[3], and to a lesser extent that they contain a large number of "easy" (according to our criterion) datasets.

We believe that our benchmarks present some advantages compared to these previous benchmarks:

- Our benchmark is clearly about *tabular* data, and we restrict our datasets to ones which exhibit clear tabular characteristics, by removing datasets without heterogeneous columns, or deterministic game-based datasets. This allows to interpret performance on our benchmark as performance on typical tabular datasets, and to leverage our benchmark to study inductive biases well-suited to tabular data. While previous benchmarks don't necessarily contain a lot of non-heterogeneous datasets like images, we think that not having a benchmark clearly about tabular data discourage its use by tabular data researchers, as it adds an unfortunate degree of freedom in the choice of which datasets to remove.

- We strive to have a more homogeneous number of samples between our different datasets (3K-10K). We believe this makes our benchmark easier to interpret, and allows to give a more definitive answer in a common setting. Furthermore, this prevent datasets with a small test set from making the evaluation noisier.

- The way we assess the difficulty of datasets is stricter than for previous benchmark, as we try to account for the different of different datasets, by removing datasets where sophisticated algorithms (Resnet and GradientBoostingTree) do not significantly outperform Logistic Regression. As tree-based methods have been shown to be superior to Logistic Regression [Fernández-Delgado et al., 2014] in our setting, a close score for these two types of models indicates that we might already be close to the best achievable score. This allows us to remove datasets which are too noisy to differentiate between two learning algorithms. All in all, this allows us to remove a lot of datasets which would not contribute meaningfully to our comparison, and in turn makes our benchmark much faster to run for other researchers.

- Despite being stricter in our datasets selection than previous benchmarks, we still manage to gather 45 datasets (which are large enough to have a precise evaluation on), by looking for datasets across various sources.

### A.6.1 OpenML CC18

We start by highlighting a few important difference differences between the CC18 benchmark and ours. More information on CC18 criteria are available at `https://docs.openml.org/benchmark/`.

- The difficulty of the dataset (used to remove too easy datasets) is not assessed in the same way: the creators of CC18 remove datasets for which a decision tree can reach 100% accuracy, while we remove datasets where sophisticated algorithms (Resnet and GradientBoostingTree) significantly outperforms Logistic Regression.

- CC-18 only contains classification datasets.

- Constraints on dataset size are less strict for CC18 (500 and 100000)

- There are some images datasets in CC18, while we only select datasets with heterogeneous features.

- Deterministic datasets (e.g game-based datasets like 'poker') are allowed in CC18.

- CC-18 contains multi-class classification problems.

---

[3]Note that if a dataset is too small, we only report the rejection criterion as 'Too small', while other criteria might also apply

Comparing our classification benchmarks to CC18, we find that 12 / 15 datasets for our numerical benchmark and 6 / 7 datasets for our categorical benchmark are not in CC18. Indeed, only 3 / 72 datasets from CC18 are used in our benchmarks. Below we show the reasons we didn't select CC18 datasets.

| Criterion | count |
| --- | --- |
| Deterministic | 2 |
| High dimensional | 1 |
| Not enough features | 1 |
| Not heterogeneous | 3 |
| Too easy | 12 |
| Too small after preprocessing | 11 |
| Too small before processing | 39 |
| Used | 3 |

### A.6.2 AutoML

We start by highlighting a few important difference differences between the AutoML benchmark and ours. More information on AutoML criteria are available at the archived link `https://web.archive.org/web/20200916061348/https://openml.github.io/automlbenchmark/benchmark_datasets.html`, but the criteria for dataset selection are less clear than for CC18.

- It is not clear if there is an objective criterion for dataset difficulty for the AutoML benchmark, as they strive to remove easy datasets, but empirically a lot of datasets from AutoML benchmarks are "too easy" according to our criterion.
- Constraints on dataset size are less strict for AutoML.
- While the creators of AutoML try to limit the number of image datasets, there are some images datasets, while we only select datasets with heterogeneous features.
- While the creators of AutoML try to limit the number of artificial datasets, there are some artificial datasets.
- The AutoML benchmark contains multi-class classification problems.

To compare our set of datasets to those of AutoML, we use the up-to-date benchmark for classification (benchmark 271 on OpenML) and regression (benchmark 269 on OpenML).

**Classification**

4 / 71 datasets from the AutoML classification benchmark (full updated version) are used in our classification benchmarks, which means 11 / 15 datasets from our numerical classification benchmark and 7 / 7 datasets from our categorical classification come from other sources. Below we show the reasons we didn't select AutoML datasets.

| Criterion | count |
| --- | --- |
| Artificial | 4 |
| Deterministic | 2 |
| Not enough features | 3 |
| Not heterogeneous | 1 |
| Other | 1 |
| Too easy | 22 |
| Too small after preprocessing | 11 |
| Too small before processing | 20 |
| Used | 4 |

**Regression**

8 / 32 datasets from the AutoML regression benchmark are used in our regression benchmarks, which means 8 / 19 datasets from our numerical regression benchmark and 4 / 13 datasets from our categorical classification come from other sources. Below we show the reasons we didn't select AutoML datasets.

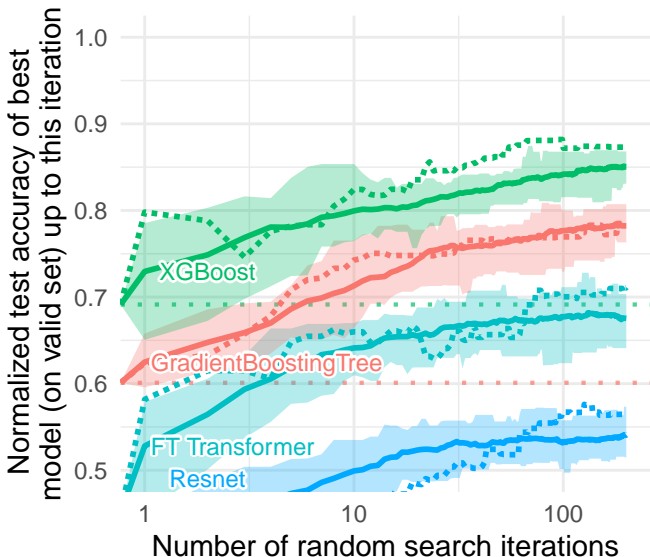

Figure 33: Comparison of random search (filled line) with **Bayesian optimization** (dotted line) on the classification on numerical features benchmark. Horizontal dotted lines correspond to default hyperparameters, which is the first step for both random search and Bayesian optimization.

| Criterion | count |
| --- | --- |
| High dimensional | 1 |
| Not enough features | 3 |
| Other | 1 |
| Too easy | 6 |
| Too small after preprocessing | 2 |
| Too small before processing | 9 |
| Used | 8 |

## A.7 Bayesian optimization

In this section, we evaluate if using Bayesian optimization instead of random search would change our conclusions. To this aim, we use Weight and Biases' Bayesian optimization algorithm (based on Gaussian processes with Matern kernels, `https://github.com/wandb/sweeps/blob/master/src/sweeps/bayes_search.py`), and run our benchmarks on numerical features with this method for 200 steps. In Figures 33 and 34, we plot the result of this approach, compared to the random search approach described in 3.3. Note that using Bayesian optimization doesn't allow us to reshuffle evaluation steps, so we only display error bars for random search.

**Results**

- Using Bayesian optimization doesn't seem to change the ordering of the different models.

- Bayesian optimization does not seem to provide a significant improvement over random search. We emphasize, however, that our experiment is a simple check that using Bayesian Optimization doesn't change our conclusions, and a proper evaluation of Bayesian optimization in this context would require running several runs of Bayesian optimization for each model and each dataset.

## A.8 Discussion on Kadra et al. [2021b]

We observed that tree-based models are superior for every random search budget, and the performance gap stays wide even after a large number of random search iteration. However, this might no longer be true when adding additional regularization techniques to our random search, such as data

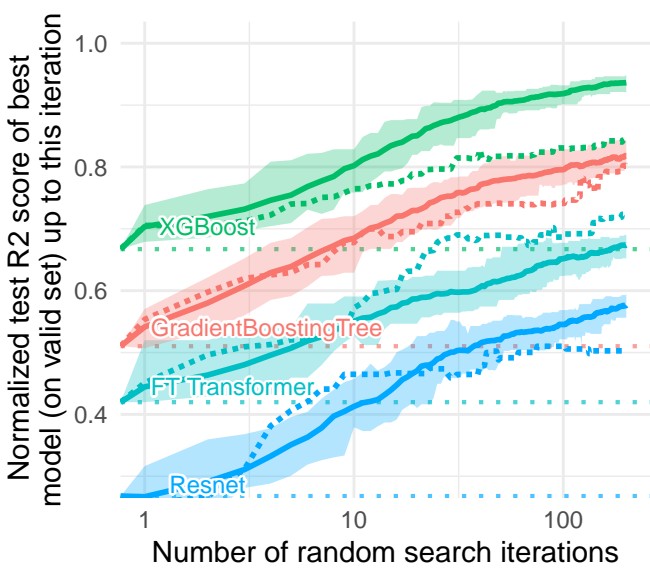

Figure 34: Comparison of random search (filled line) with **Bayesian optimization** (dotted line) on the regression on numerical features benchmark. Horizontal dotted lines correspond to default hyperparameters, which is the first step for both random search and Bayesian optimization.

augmentation. Indeed, Kadra et al. [2021a] find that searching through a "cocktail" of regularization on a Multi-Layer-Perceptron is competitive with XGBoost after half an hour of tuning (for both models), though the datasets considered in their paper are quite different, in particular with the presence of "deterministic" game-inspired datasets in Kadra et al. [2021b], on which their method performs very well and contributes markedly to the overall benchmark results.

### A.9 Which hyperparameters perform well on tabular data?

Our random search results provide insights into which hyperparameters are important for learning on tabular data. Below we present a measure of hyperparameters importance in our classification benchmark on numerical features.

**Methodology**  Accuracy is normalized (see 3), and negative scores are truncated to zero. For each model, we fit a RandomForest classifier with default hyperparameters to predict these scores from the model's hyperparameters and the dataset, and we compute the feature importance for each of the hyperparameter ("rf_importance"). This gives us a score which represent how much an hyperparameter should be tuned. To measure if an hyperparameter has a positive or negative impact on the performance, we also compute the coefficient of a LinearRegression trained on the same task that the RandomForest ("lin_coef")

**Results**  The learning rate is by far the most important parameter for neural networks and gradient-boosted trees. Note that the linear coefficient is not always very high, which suggests that the learning rate should be tuned for each dataset. For tree-based models, the depth of the trees is another very important parameter, and it seems that deeper trees help, even for gradient-boosted trees. This observation is related to our "Finding 1" 5.2, as deeper trees enable very irregular patterns to be learned.

Below we give the results for each architecture.

**MLP**

| names | rf_importance | lin_coef |
|---|---|---|
| lr | 0.22 | -0.04 |
| d_layers | 0.18 | -0.01 |
| d_embedding | 0.15 | 0.0 |
| n_layers | 0.14 | -0.1 |
| batch_size | 0.03 | -0.01 |
| lr_scheduler | 0.01 | 0.0 |

## Resnet

| names | rf_importance | lin_coef |
|---|---|---|
| lr | 0.08 | 0.01 |
| normalization_layernorm | 0.05 | 0.0 |
| n_layers | 0.04 | -0.03 |
| d | 0.02 | 0.0 |
| hidden_dropout | 0.02 | 0.01 |
| residual_dropout | 0.01 | 0.0 |
| batch_size | 0.01 | 0.0 |
| optimizer__weight_decay | 0.0 | 0.0 |
| d_hidden_factor | 0.0 | 0.0 |
| d_embedding | 0.0 | 0.0 |
| lr_scheduler | 0.0 | 0.0 |

## FT Transformer

| names | rf_importance | lin_coef |
|---|---|---|
| lr | 0.06 | 0.0 |
| residual_dropout | 0.04 | -0.02 |
| d_token | 0.03 | 0.0 |
| n_layers | 0.02 | 0.0 |
| ffn_dropout | 0.02 | 0.0 |
| d_ffn_factor | 0.01 | 0.0 |
| kv_compression | 0.01 | -0.01 |
| optimizer__weight_decay | 0.01 | 0.0 |
| attention_dropout | 0.01 | 0.0 |
| lr_scheduler | 0.0 | 0.0 |
| kv_compression_sharing_key-value | 0.0 | 0.0 |
| batch_size | 0.0 | 0.0 |

## SAINT

| names | rf_importance | lin_coef |
|---|---|---|
| lr | 0.28 | -0.11 |
| depth | 0.15 | -0.07 |
| dim | 0.05 | -0.03 |
| dropout | 0.03 | 0.0 |
| heads | 0.01 | 0.0 |
| batch_size | 0.01 | 0.0 |
| val_batch_size | 0.0 | 0.0 |

## XGBoost

| names | rf_importance | lin_coef |
|---|---|---|
| learning_rate | 0.26 | 0.01 |
| min_child_weight | 0.07 | -0.04 |
| max_depth_2.0 | 0.05 | 0.13 |
| reg_alpha | 0.05 | -0.03 |
| n_estimators | 0.03 | 0.01 |
| max_depth_4.0 | 0.02 | 0.27 |
| subsample | 0.02 | 0.01 |
| colsample_bytree | 0.02 | 0.0 |
| max_depth_3.0 | 0.02 | 0.21 |
| reg_lambda | 0.01 | 0.0 |
| gamma | 0.01 | 0.0 |
| max_depth_10.0 | 0.01 | 0.35 |
| max_depth_11.0 | 0.01 | 0.35 |
| max_depth_7.0 | 0.01 | 0.33 |
| max_depth_5.0 | 0.01 | 0.3 |
| max_depth_6.0 | 0.01 | 0.32 |
| max_depth_8.0 | 0.01 | 0.34 |
| max_depth_9.0 | 0.01 | 0.34 |
| colsample_bylevel | 0.01 | 0.0 |

## GradientBoostingClassifier

| names | rf_importance | lin_coef |
|---|---|---|
| learning_rate | 0.4 | -0.02 |
| n_estimators | 0.15 | 0.09 |
| max_depth_None | 0.14 | 0.36 |
| max_depth_4.0 | 0.04 | 0.19 |
| max_depth_3.0 | 0.02 | 0.11 |
| min_samples_leaf | 0.01 | 0.0 |
| subsample | 0.01 | 0.0 |
| loss_exponential | 0.0 | 0.0 |

## RandomForest

| names | rf_importance | lin_coef |
|---|---|---|
| max_depth_None | 0.43 | 0.54 |
| max_depth_4.0 | 0.07 | 0.24 |
| max_depth_3.0 | 0.03 | 0.13 |
| min_samples_leaf | 0.01 | -0.01 |
| n_estimators | 0.01 | 0.0 |
| max_features_None | 0.01 | -0.06 |
| bootstrap | 0.01 | 0.01 |
| max_features_log2 | 0.0 | 0.04 |
| criterion_gini | 0.0 | 0.0 |
| max_features_0.2 | 0.0 | 0.04 |
| max_features_sqrt | 0.0 | 0.04 |
| max_features_0.4 | 0.0 | 0.05 |
| max_features_0.5 | 0.0 | 0.04 |
| max_features_0.6 | 0.0 | 0.04 |
| max_features_0.7 | 0.0 | 0.03 |
| max_features_0.8 | 0.0 | 0.01 |
| max_features_0.9 | 0.0 | -0.01 |
| max_features_0.3 | 0.0 | 0.05 |

### A.9.1 Should we run tree-based models longer?

In the main section of this paper, we chose a maximum number of estimators (1000) and patience (20) for tree based models such that these models are significantly faster than their neural networks counterparts. However, the maximum number of 1000 is often reached, meaning that we train these models short of convergence. In Figures 36 and 38, we compare the performance of our tree-based models with tree-based models ran with a higher number of maximum estimators (40K instead of

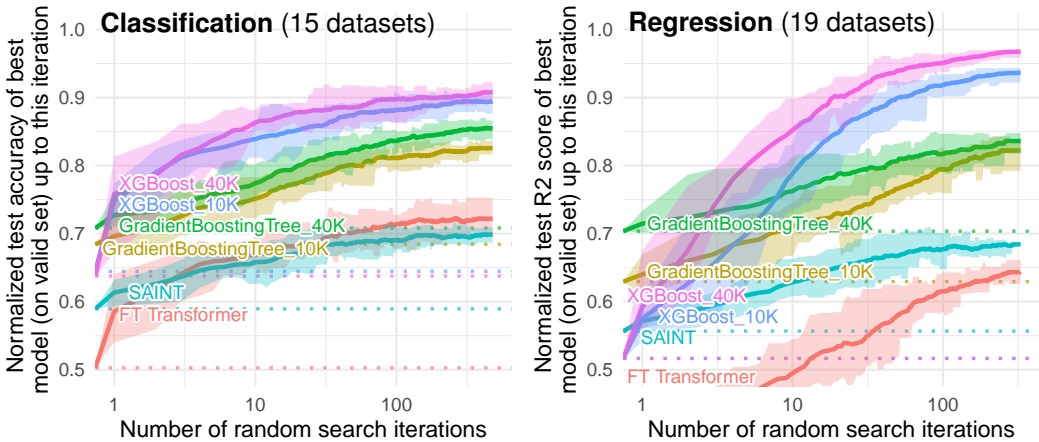

Figure 35: **Benchmark on medium-sized datasets, with only numerical features**. We compare gradient boosting models ran for a maximum number of 40K estimators and an early stopping patience of 100 with the same models ran with a maximum of 1K estimators, and a patience of 20. Note that the default setting is run with these respective parameters (40K and 10K) instead of Sklearn's default.

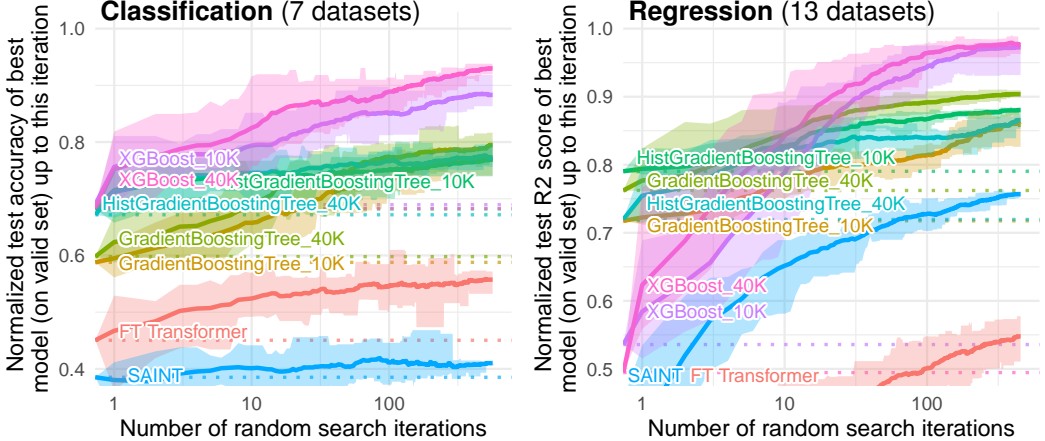

Figure 36: **Benchmark on medium-sized datasets, with both numerical and categorical features**. We compare gradient boosting models ran for a maximum number of 40K estimators and an early stopping patience of 100 with the same models ran with a maximum of 1K estimators, and a patience of 20. Note that the default setting is run with these respective parameters (40K and 10K) instead of Sklearn's default.

1K) and a higher patience (100 instead of 20). We can see that this slightly improve the per-random-search-iteration performance, as well as the best performance reached, of tree-based models (figure 36) albeit at the cost a significantly higher compute time (figure 38).

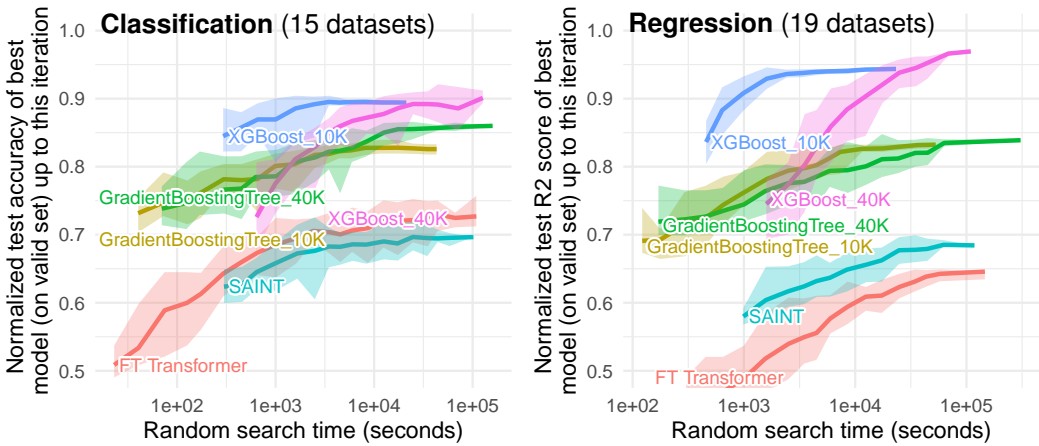

Figure 37: **Time benchmark on medium-sized datasets, with only numerical features**. We compare gradient boosting models ran for a maximum number of 40K estimators and an early stopping patience of 100 with the same models ran with a maximum of 1K estimators, and a patience of 20. Note that the default setting is run with these respective parameters (40K and 10K) instead of Sklearn's default.

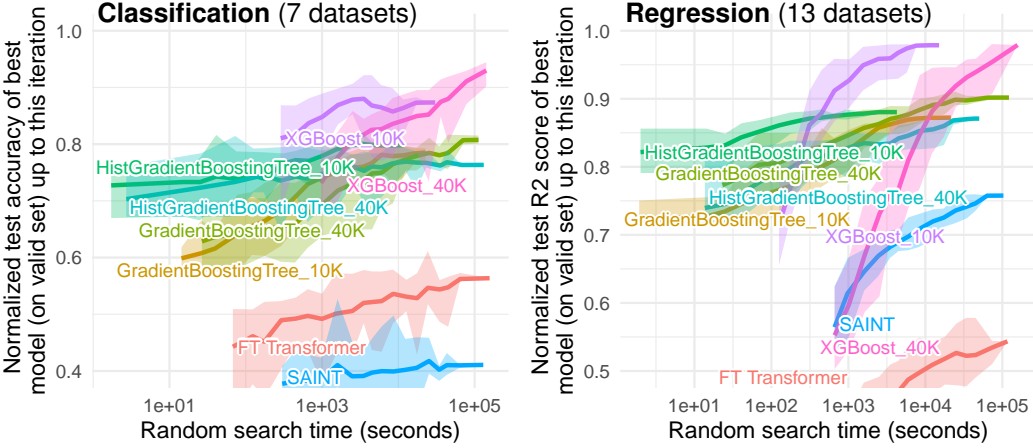

Figure 38: **Time benchmark on medium-sized datasets, with both numerical and categorical features**. We compare gradient boosting models ran for a maximum number of 40K estimators and an early stopping patience of 100 with the same models ran with a maximum of 1K estimators, and a patience of 20. Note that the default setting is run with these respective parameters (40K and 10K) instead of Sklearn's default.

### A.9.2 Should we tune the number of estimators for tree-based models?

In Figures 39 and 40, we study the usefulness of tuning the number of estimators of tree-based models (as done in Hyperopt-sklearn), by comparing two settings. In the first setting, denoted as *tune*, we tune the number of trees, and do not use early stopping. The distribution used for tuning this parameter are those from Hyperopt-sklearn, namely:

- RandomForest: LogUniform(9.5, 3000.5)
- GradientBoostingTree: LogUniform(10.5, 1000.5)
- XGBoost: Uniform(100, 6000)

In the second setting, we set the number of estimators to a high number for each model, and use early stopping for GradientBoostingTree and XGboost, with a "patience" of 20 and a default tolerance, and a maximum number of 1000 estimators. For RandomForest, we do not use early stopping and set the number of estimators to 250.

**Results** The results we obtain in both settings seem relatively similar, and very close for XGboost. Tuning the number of estimators seems to harm the random search performance for Random Forests a little but improve it for the "best reached" prediction for GradientBoostingTrees (though it harm performance with a small hyperparameter search budget). We find this latter observation surprising, and would like to investigate further. Note that this observation does not disappear when using a maximal number of estimators of 40K (instead of 1K) and a patience of 100 (instead of 20).

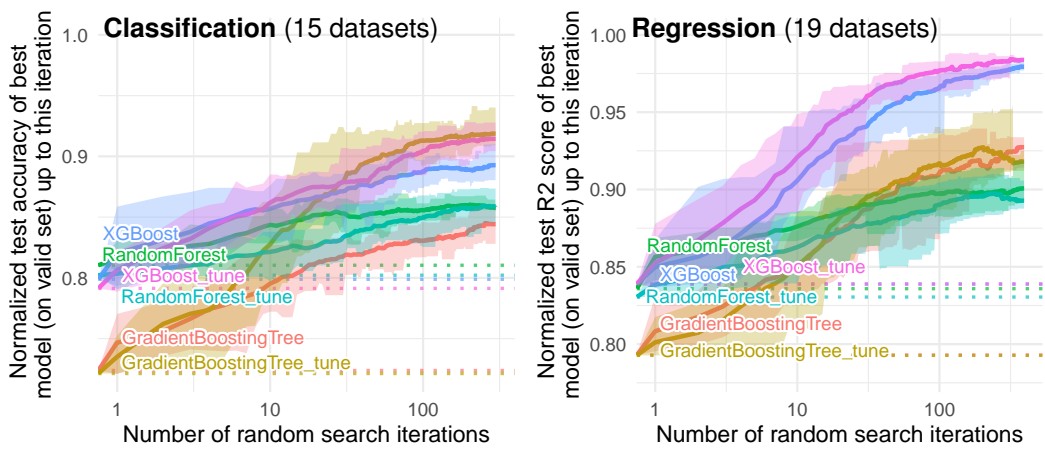

Figure 39: Comparison of random searches for tree-based models when the number of estimators is fixed or tuned, for numerical features only.

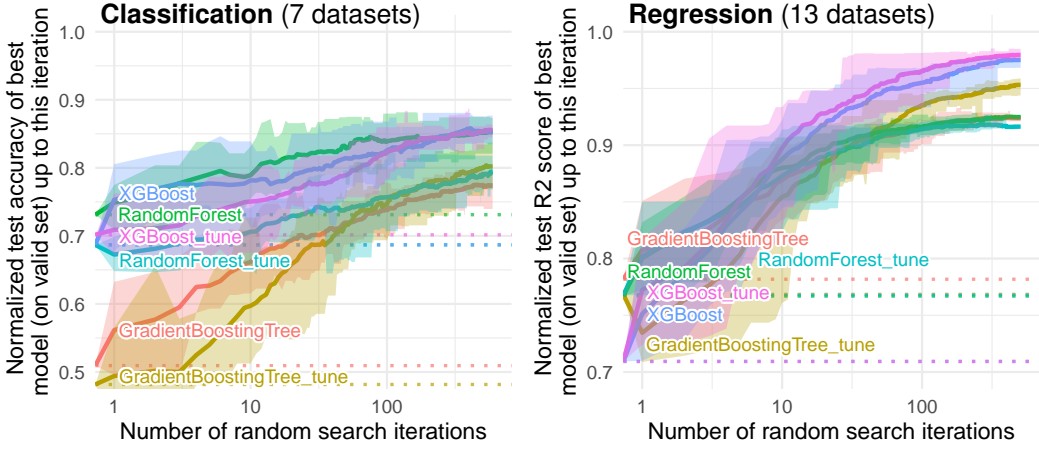

Figure 40: Comparison of random searches for tree-based models when the number of estimators is fixed or tuned, for both numerical and categorical features.

### A.10 How to use our benchmark?

All instructions to use our benchmark and reproduce our results are available at our repository: `https://github.com/LeoGrin/tabular-benchmark`. To ease the use of our benchmark and the reproducibility of our results we provide:

- The selected and transformed datasets as an OpenML suite. All links to the transformed and original datasets are also in A.1.

- A CSV file containing the results of all our random searches. It can be used to cheaply benchmark a new method.

- The code used to produce our benchmark and our experiments.