# OpenReview forum: "Why do tree-based models still outperform deep learning on typical tabular data?"
_NeurIPS.cc/2022/Track/Datasets_and_Benchmarks — NeurIPS 2022 Datasets and Benchmarks _

### Official Review · Reviewer_w58A · 2022-07-22

**Rating:** 4
**Confidence:** 4
**Clarity:** The paper is well written and can be …

**Strengths:**

* Tabular data are still highly relevant and so is the question of which type of model to use and whether neural networks are worth it
* Datasets part of the benchmark are made accessible through re-uploads or existing uploads via OpenML; code is provided via a public GitHub repository
* The most interesting contribution of the paper is in my opinion neither the new benchmark, nor the result that tree-based methods outperform neural networks (which was already reported in some prior work - which the authors correctly cite), but rather the experimental investigation of the inductive bias of tree-based models compared to neural networks (Section 5)
  - The experimental design is convincing and the conclusions (neural networks being biased towards smooth solutions, MLP-like networks not being robust to uninformative features and rotation invariance hindering performance on tabular data) are sensible and mostly go in line with hypotheses (and findings) of prior work
  - Section 5 therefore provides a testable hypothesis to the research community regarding the design of future neural networks for tabular data (i.e., a neural network that can learn irregular representations, is robust to uninformative features and not rotation invariant should show strong performance)


**Weaknesses:**

* Overall, the paper reads much more like an empirical investigation rather than a presentation of a new tabular benchmark; in essence, the paper provides both (by performing the empirical investigation on the new tabular benchmark); but it is unclear of what additional benefit this new collection of existing datasets will be, i.e., should all new approaches for supervised learning on tabular data now be benchmarked on this new collection? If yes, one may argue that the current collection may be too homogeneous, due to the preprocessing and all the restrictions listed in 3.1
* Also, the need for coming up with a new collection of tabular datasets is not thoroughly justified, i.e., the authors mention both OpenML CC-18 (72 classification tasks) and the automl benchmark (39 classification tasks) but state that ```[they] contain tabular data, but also include images and artificial datasets, which may explain why they have not been used in tabular deep learning papers```, however, the majority of tasks in the CC-18 and automl benchmark are based on standard tabular datasets (with some exceptions like ```fashion-mnist```); also a question I had in mind when reading the paper was: Would main results and conclusions have looked different if benchmarks were performed on CC-18 or the automl benchmark instead of the new collection? Probably not, due to overlap of datasets being used in all three benchmarks (?)
* For hyperparameter tuning, a random search with a budget of around 400 iterations was used for each learner (defaults used as the first iteration). While this allows for a ``fair'' comparison, it could also be the case that the HPO problem is much more difficult for neural networks than for tree-based models (where HPO loss landscapes may be more benign). If this would be the case, part of the results could also be explained by not using a sophisticated optimizer, i.e., it could be the case that Bayesian Optimization can identify much more potent configurations of the neural networks which could allow for closing the performance gap (partially). All in all, this is only speculation, but could be investigated in a small ablation study.
* While I do find the empirical investigation regarding the inductive bias of tree-based models and neural networks interesting (Section 5), the analyses were mostly performed visually, e.g., by inspecting box plots. The experimental design, however, would, in principle, allow for more detailed statistical analyses, e.g., modeling the test score as a function of the learner and length scale (of the smoothing; in Section 5.1) and testing for significance of effects

Some additional (mostly minor) remarks:
* In the abstract and main paper it is stated that `45` datasets are part of the benchmark. However, in the Appendix, in total `55` datasets are described: numerical classification 15, numerical regression 20, categorical classification 7, categorical regression 13 (this is also in contrast to the numbers listed in the figures, i.e, Figure 1: 15, 19, 7, 14
* It may be helpful to discuss why one-hot-encoding of categorical features was employed (instead of, e.g., impact encoding or entity embedding with the latter one more frequently being used in neural networks for tabular data (Guo & Berkhahn, 2016))

All in all, my rating is mostly determined by the fact that I do not see much benefit of the new benchmark collection itself (compared to CC-18 or the automl benchmark) although I do find the empirical investigation interesting.

**Additional Feedback:**

When trying to run ```python data/download_data.py``` (after cloning the GitHub repository) I get the following error (probably due to some directories not being created on the fly):
  ```
  Saving datasets from suite: numerical_regression
Downloading dataset cpu_act
Traceback (most recent call last):
  File "/tmp/tabular-benchmark/data/download_data.py", line 29, in <module>
    save_suite(suites_id["numerical_regression"],
  File "/tmp/tabular-benchmark/data/download_data.py", line 17, in save_suite
    with open("{}/data_{}".format(dir_name, dataset.name), "wb") as f:
FileNotFoundError: [Errno 2] No such file or directory: 'data/numerical_only/regression/data_cpu_act'
  ```

**Correctness:**

The benchmark collection is constructed in a sound way, although quite homogeneous (on purpose).
The experimental design is appropriate.
Evaluation is mostly performed visually - please see also my comment in the Weaknesses Section above.

**Documentation:**

Data collection and preprocessing is described in detail; code is also available.
Final datasets are made available through (re-uploads or existing uploads) via OpenML.

**Ethics:**

Not applicable.

**Relation To Prior Work:**

Prior work is discussed clearly.
The own contribution is stated clearly.

**Summary And Contributions:**

This paper aims to answer the question of why tree-based models outperform neural networks on tabular data.
Contributions are:
* A new collection of 45 (already public available) tabular datasets (classification and regression tasks) that have been (partially) preprocessed to establish a homogeneous collection with ``clear characteristics'' of tabular data
* A benchmark study comparing tree-based models (XGBoost, random forest, gradient boosted trees) and neural networks for tabular data (MLP, Resnet, FT Transformer, SAINT) on this new benchmark; results show that tree-based models overall outperform neural networks
* An empirical investigation of the inductive bias of tree-based models, contrasted with that of neural networks for tabular data, which serves as a basis for answering the question of why tree-based models outperform neural networks
* (Raw data as obtained from performing the benchmark, including the HPO runs)

---

> ### Author Response · Authors · 2022-08-24
> **Answer**
>
> We thank the reviewer for the insightful review, and we now answer the questions:
>
> ## Contributions
>
> Our contribution is threefold: 1) we define a standard set of datasets with strict “tabular” properties 2) we evaluate new models on this set of datasets, taking into account HPO and sharing our compute-intensive results 3) we delve into the performance gap between tree-based models and neural networks, and shed light on their different inductive biases. The third contribution is only possible because of the clear-cut definition of a benchmark suite. Indeed, an heterogeneous benchmark would have created new sources of performance difference, washing out the signal of the empirical study.
>
> ## Should all new approaches for supervised learning on tabular data now be benchmarked on this new collection?
>
> We do think that it would be a good thing! Indeed, even if our benchmark does not include some crucial challenges for tabular data (such as missing data), performing well on our benchmark is probably a first step to have a good algorithm for tabular data. While new approaches should also be evaluated in more challenging conditions, we think that it might be best to separately develop inductive biases well-suited for tabular data in the strictest sense (i.e well-suited for our benchmark), and methods made to deal with missing data or high-cardinality variables.
>
> ## Comparison with previous benchmarks
>
> We have added a new section A.6 comparing our benchmark to the CC-18 and the AutoML benchmarks. While these benchmarks' selection procedures are quite similar to ours, we highlight a few important differences. Crucially, these small differences in the selection procedure make for very large differences in the datasets selected: for instance, only 3 / 72 datasets from CC-18 are used in our benchmarks. The main reason is that previous benchmarks include many small-scale datasets, and a significant number of "easy" datasets.
>
> We believe that our benchmark presents several advantages which justify its usefulness:
> - We choose only datasets which exhibit clear tabular characteristics. This allows us to measure performance on typical tabular datasets, and to leverage our benchmark to study inductive biases well-suited to tabular data. While previous benchmarks don't necessarily contain a lot of non-heterogeneous datasets like images, we think that not having a benchmark clearly about tabular data discourages its use by tabular data researchers, as it adds an unfortunate degree of freedom in the choice of which datasets to remove.
> - We strive to have a more homogeneous number of samples between our different datasets (3K-10K). This allows us to give a more definitive answer in a common setting, and prevents datasets with a small test set from making the evaluation noisier.
> - The way we assess the difficulty of datasets is more stringent, as we try to account for the different Bayes rates of different datasets, by removing datasets where sophisticated algorithms do not significantly outperform Logistic Regression. This allows us to remove a lot of datasets which would not contribute meaningfully to our comparison (such as noisy datasets), and in turn makes our benchmark much faster to run for other researchers.
> - Despite being stricter in our datasets selection, we still manage to gather 45 datasets (large enough to have a precise evaluation on), by looking across various sources.
> - CC-18 only contains classification tasks.
>
> ## Bayesian optimization
>
> We have added a new section A.7, where we try Bayesian optimization (using “Weights and Biases” implementation, a very popular tool in deep learning) and find that:
> - Using BO does not seem to change the ordering of the different models
> - BO seems to underperform random search for most models in our settings. This under-performance seems more severe for neural networks, where BO struggles to find the best hyperparameters.
>
> ## Statistical analysis
>
> In A.4, we have added a new paragraph “Quantitative analysis” for each experiment. The quantitative results match our plots and our conclusions, and are all statistically significant.
>
>
> ## Bug
>
> We apologize for this, and have fixed this issue.
>
> ## Discrepancy in the reported number of datasets
>
> This is due to some datasets being used in several benchmarks. For instance certain datasets are used with their categorical features in the categorical benchmark, and without them in the numerical benchmarks.
>
> ## Categorical encoding
>
> One-hot encoding is used for GradientBoostingTrees, RandomForest and XGBoost. For FT Transformer and SAINT, we use the specific categorical encodings described in the original papers. For Resnets and MLPs, we embed the categorical features as described in [1]. We also use HistGradientBoosting with its native categorical feature support for comparison. All in all, we think that this reflects standard practice and allows for a fair comparison.
>
> [1]: Revisiting deep learning for tabular data.

---

> > ### Comment · Reviewer_w58A · 2022-09-01
> > **Reply to the authors ii**
> >
> > I sincerely apologize to the authors for stating this criticism in such a late stage of the reviewing process, however, I must admit that only when thinking about the poor performance of BO (and those parts were only included very recently) I also started to think much more detailed about the (at first glance sound) experimental setup in general.
> > Given the current experimental setup and evaluation protocol, I find it difficult to recommend the paper for acceptance.
> >
> > I also performed a small investigation to backup my claim that the mean and variance values based on shuffles are not directly related to the actual measures of interest (expected performance and uncertainty).
> > To do so, I ran a Random Search for 400 iterations on the Hartmann4D function.
> > I then shuffled that Random Search run 15 times and calculated the mean of the best y and the standard error of the mean over the 400 iterations (rs_shuffle_1).
> > Using a different random seed, I reran the above procedure resulting in rs_shuffle_2.
> > Finally, I ran the Random Search itself 15 times with a different random seed and calculated the mean and the standard error of the mean over those 15 statistical replications (rs_15).
> >
> > I performed the whole procedure two times and you can see the results in the Figure I attached (left plot first replicate, right plot second replicate).
> > Note that the performance of the Random Search using actual statistical replications (the red line) does not differ that much between the two replicates (and the difference would go close to zero if I had used more replications).
> > However, looking at the shuffled Random Search versions (green and blue), the performance outcome can be quite different and is related to good / back luck with respect to the random seed.
> >
> > [Figure](https://imgur.com/a/qm6BEOQ)
> >
> > [Kadra et al. 2021](https://arxiv.org/abs/2106.11189)
> >
> > [Eggensperger et al. 2021](https://arxiv.org/abs/2109.06716)

---

> > > ### Author Response · Authors · 2022-09-02
> > > **Answer**
> > >
> > > We thank the reviewer for engaging with our paper, and for providing experiments.
> > >
> > > # Bayesian optimisation
> > > We have double checked our analysis, and found a small mistake. Without this mistake, it is not clear if Bayesian Optimization underperforms random search in our case. What is still clear, however, is that Bayesian optimization doesn’t change the ordering of the models. We have updated the manuscript. Note that other studies have found that in some settings  Bayesian Optimization did not seem to outperform Random Search (see Bouthillier et al, “Accounting for Variance in Machine Learning Benchmarks”, fig F.2, which is a study ran by experts of hyper-parameter selection)
> > >
> > > > comparing a single BO run to an (in essence) single Random Search run
> > >
> > > Our comparison of random search and Bayesian optimization is not meant to be more than a check that 1 run of Bayesian Optimization does not behave too unexpectedly compared to the distribution of random search (resampled) runs. For a proper comparison, we would indeed need to run several Bayesian Optimisation runs.
> > >
> > > # Criticism of our method
> > >
> > > We agree with the reviewer than using independent random search runs would provide better estimates, but we think we’re justified in using our method:
> > > - We *are* limited by compute, especially for deep learning models, and we want our procedure to be easily reproducible by compute-limited researchers, otherwise it won’t be adopted.
> > > - In a new complementary experiment, we have run our classification benchmark on numerical features with 15 independent random search runs on a GradientBoostingTree (in blue), and compared the results to our method (in red). In practice, the two procedures give very similar results: https://imgur.com/NAO5tea This also suggests that 400 steps of random search is enough to meaningfully explore the hyper-parameters space in our case.
> > > - Our method samples a single hyper-parameter optimization run and performs an imperfect simulation of multiple runs (see below) to get a better estimate of the distribution over multiple runs for a given dataset. Yet, while sampling a single run can lead to un-accounted variance, it is not biased and as we compare models across many datasets, this variance will average out. Thus, the order of the different curves is well estimated.
> > >
> > > Furthermore, it is important to distinguish two regimes of the curve. On the left-part of the curve, only a small number of random-search points are used, e.g. drawing many times 10 points randomly out of 400. As these are independent, they approximate well drawing many times 10 points completely at random. However, on the other end of the curve, the right most part, 399 random-search points may be used out of 400. In which case, the ensembles drawn are always almost identical, and the uncertainty estimation may not be representative of independent runs of random search (though in practice, the difference seems small). However, we agree that this subtle nuance between the two ends of the curve can be misleading, and we have added a clarification in Section 4.2.

---

> > ### Comment · Reviewer_w58A · 2022-09-01
> > **Reply to the authors i**
> >
> > I thank the authors for the revision of the paper and for answering my questions.
> >
> > I believe the paper benefits from the clarification of the contribution as well as the comparison to previous benchmarks.
> >
> > I also think that the additional statistical analyses are helpful.
> >
> > Regarding the Bayesian Optimization (BO) section:
> > For me, it is highly unexpected that BO performs consistently poorly compared to a simple Random Search (this finding would go against most findings in the HPO literature in the last years, see, e.g., the experimental study in the HPOBench paper [Eggensperger et al. 2021]).
> > Initially, I believed this may be simply caused by a poorly performing BO implementation (or some bug or wrong specification), but still you would expect BO to be at least within the uncertainty estimates of the Random Search - which is not the case for the majority of the learners.
> > However, after re-reading the section, as well as the experimental setup, I believe this is the result of comparing a single BO run to an (in essence) single Random Search run (given a certain learner and a certain dataset).
> > After all, performance of Random Search (and also BO) can strongly depend on the random seed and if no replications are performed (over random seeds), results may be strongly influenced based on bad / good luck with respect to the random seed.
> >
> > This also made me question the general experimental setup. Below, I will outline what I mean by ''an (in essence) single Random Search run''.
> > If I understand it correctly:
> >   1) For each learner and dataset a single run of Random Search was performed (in general, for 400 iterations)
> >   2) To construct something like a statistical replicate, the evaluation order of that Random Search run is permutated or ''shuffled'' 15 times. Based on these 15 ''shuffles'' a mean and variance estimate of the performance can be calculated for each time step.
> >      * a) I would argue, however, that these mean and variance values are no estimates of the actual measures of interest, which are the expected performance and its uncertainty (where the expectation should be taken over different random seeds and / or different train / validation / test splits). Especially as the random search iterations are between 100 and 400 which is not enough to cover the search spaces in the paper (i.e., if a very large budget, say 10^6 iterations on small search spaces would have been used - not that this would have been possible here- I would argue that shuffling or bootstrap like estimates are not that problematic).
> >      * b) It would have been desirable to perform actual statistical replications and given the homogeneous, rather small datasets, runtime should not have been that much of a limiting factor.
> >      * c) I believe that many arguments made in the paper with respect to the tuning results lack proper empirical support due to effectively only considering a single Random Search run, being subjective to the random seed.
> >   3) Each shuffle of a Random Search run for a given learner and dataset is normalized given the dataset and performance of the other learners (as described in 3.3)
> > 	  * a) Naturally, this normalization itself then again strongly depends on the random seeds as the top and worst performance are determined based on single runs of random search on a given learner and dataset and not actual statistical replications.
> >
> > Therefore, I believe that the mean performance and uncertainty estimates shown in the main Figures (1-6) may be misleading and also do not represent what one would expect (i.e., one would expect to see the expected performance and its uncertainty over replications and not the performance for a single run with uncertainty estimated via reshuffling) and the conclusions one may draw based on the plots about, e.g., the ranking of learners with respect to their final performance and the uncertainty may be potentially misleading.
> >
> > Although it is not uncommon to base conclusions on only a single run for each learner and dataset combination, I would argue that in this case a statistical analysis in the form of, e.g., a Friedman Test on the final performance of each learner x dataset pair and follow-up Nemenyi Tests (i.e., critical difference plots) are needed to be able to properly rank learners with respect to their performance (see, e.g., the analyses performed in Kadra et al. 2021).
> >
> > Based on the current experimental setup (shuffles instead of actual statistical replications and normalized anytime performance plots based on these shuffles), I find it difficult to evaluate whether the empirical results actually support the claims made in the paper and if so to which extent.
> > (Not that I believe that the claims may not be true but rather that it is difficult to assess their validity based on the current evaluation protocol and the plots provided).

---

### Official Review · Reviewer_rCUL · 2022-07-24
**Why do tree-based models still outperform deep learning on tabular data?  - Review**

**Rating:** 6
**Confidence:** 3
**Correctness:** Everything seems correct to me

**Strengths:**

The research work is novel and its potential impact is evident. It not only provides a unprecedentedly large and publicly available dataset and benchmarks, but also reveals the inductive biases of tree-based models and neural networks that could influence their performance on tabular data.

The paper is written in an organized manner, with detailed illustrations to aid the audience to understand the results. The authors made good use of figures and plots to highlight trends and point out their observations, especially in Section 5.

The authors build upon their findings and explain the limitations to their work and provides suggestions on how future research involving tabular data could be improved.

The dataset is publicly available, and this work is the first standard benchmark for tabular data that follows several clearly defined criteria. Results of their findings are presented in graphs with clear figure descriptions. The authors also mentioned limitations and future research directions.

**Weaknesses:**

The empirical evaluation (the main strength of the paper) is not comprehensive enough and may deserve more effort to find more about this difference in behavior. Additionally, although the authors indicated that they are the first to do so, they haven't argued about why it is that important and how future research might benefit from such empirical study.

There are no benchmarked tasks on the dataset. Just “Classification” and “Regression” in Figures 1 and 2 are not clear to readers.

The authors are encouraged to make a table to compare their dataset and the other related work mentioned in section 2. The authors should also consider including more details on their benchmark and dataset (such as hyperparameters, train-test splits) in the appendix section.

**Additional Feedback:**

It would also be interesting to see the performance of tree-based models and NNs on large datasets. When the authors include this in the appendix, they should consider explaining whether the inductive biases that make tree-models favourable for tabular data still holds true for large datasets.

Out of curiosity, why does having images and artificial datasets in addition to tabular data prevent some benchmarks from being used in tabular deep learning? Do they only have to include tabular data so that we can use them in benchmarks extensively? It seems like you recommend so in the third paragraph of section 2 and I want to hear more about that, especially since this sentence is not cited.

**Clarity:**

While the paper is written in an organized fashion, it has some minor language issues. I encourage the authors to make some syntactical revisions.

**Documentation:**

The dataset and other supplementary materials are publicly available through openML with clear intended use. The authors should include as much information as possible about the benchmark in the appendix section.

**Relation To Prior Work:**

The paper thoroughly discussed previous related work and highlighted why their research work is needed.

**Summary And Contributions:**

The authors aim to compare deep learning models and tree based models on tabular data. They contributed an extensive benchmark consisting of both tree-based learners and neural networks, which is run on a set of 45 datasets from various domains. They showed that tree-based models remain state-of-the-art on medium sized tabular data and proposed explanations for this phenomenon. They also highlighted desirable biases for tabular data to guide future researchers.

---

> ### Author Response · Authors · 2022-08-24
> **Answer**
>
> We thank the reviewer for his positive review, and we now answer the questions:
>
>
> > The empirical evaluation (the main strength of the paper) is not comprehensive enough and may deserve more effort to find more about this difference in behavior. Additionally, although the authors indicated that they are the first to do so, they haven't argued about why it is that important and how future research might benefit from such empirical study.
>
> We believe that shedding lights on the different inductive biases of tree-based models and neural networks can guide researchers trying to conceive new architectures and learning methods tailored to tabular data. In turn, well-performing neural networks on tabular data, through their differentiability, could be composed or trained jointly with other deep learning blocks, and enable new applications like self-supervised or adversarial training, for data generation or transfer learning.
>
> > The authors are encouraged to make a table to compare their dataset and the other related work mentioned in section 2
>
> We have added a new section in the Appendix (A.6) comparing our benchmark to previous ones (CC-18 and the AutoML benchmark), and showing how much overlap there is in the form of several tables. While these benchmarks' selection procedures are quite similar to ours, we highlight a few important differences. Crucially, these small differences in the selection procedure make for very large differences in the set of datasets selected: for instance, only 3 out of 72 datasets from CC-18 are used in our benchmarks. Looking into the reasons for these differences, we find that the biggest source of difference between the CC-18 and AutoML benchmarks and ours is that these include many small-scale datasets, and to a lesser extent that they contain a large number of "easy" (according to our criterion) datasets. In this Appendix section A.6, we also discuss the advantages of our benchmark in more depth.
>
> > Out of curiosity, why does having images and artificial datasets in addition to tabular data prevent some benchmarks from being used in tabular deep learning? Do they only have to include tabular data so that we can use them in benchmarks extensively?
>
> While we can only speculate on the reason why benchmarks like CC-18 are not widely used in articles on tabular data deep learning, we think that not having a benchmark clearly about tabular data discourages its use by tabular data researchers, as it adds an unfortunate degree of freedom in the choice of which datasets to remove. In addition, we think that such benchmarks contain too many datasets which do not differentiate meaningfully between different algorithms, which might discourage their use by compute-parsimonious researchers (see A.6).
>
> >The authors should also consider including more details on their benchmark and dataset (such as hyperparameters, train-test splits) in the appendix section.
> > The authors should include as much information as possible about the benchmark in the appendix section.
>
> Hyperparameter spaces are specified in the Appendix A.3, and specific hyperparameters chosen during the random search are available on the Github repo as a csv file. The train / test / validation sizes are specified in A.3.
> We have added the size of the validation set used for early stopping, which we had forgotten to specify in the Appendix. If the reviewer feels that we have forgotten anything else, we will be happy to add it.
>
> > It would also be interesting to see the performance of tree-based models and NNs on large datasets. When the authors include this in the appendix, they should consider explaining whether the inductive biases that make tree-models favourable for tabular data still holds true for large datasets.
>
> An initial comparison of tree-based models and neural networks on large datasets is available in the Appendix A.2.2 , but we have found it hard to gather a sufficient amount of open-source large datasets to provide a definitive answer in this setting. For this reason, studying the inductive biases of our models in this setting would probably be hazardous, and we prefer to leave this to future works.
> We have added a new section to the appendix (A.5.2) investigating which parts of our dataset filtering process remove the largest number of datasets. The conclusion is that gathering a sufficient number of large datasets would require removing a lot of restrictions at once, which would change our benchmarks a lot and could make them harder to interpret. Another solution could be to spend more time looking for open-source large-scale datasets in the wild, but this would be a very time consuming endeavor that we leave for future work. In addition, assuming that the datasets on openml reflect a wider practice, these large settings are less frequent.
>
>
> > While the paper is written in an organized fashion, it has some minor language issues.
>
> We have proof-read our paper again, and made a few changes.

---

### Official Review · Reviewer_Damk · 2022-07-27
**Deep Learning Continues To Not Be All You Need**

**Rating:** 7
**Confidence:** 5
**Correctness:** The paper appears to be correct in it…
**Clarity:** The paper is very clear and well writ…

**Strengths:**

Given the large amount of papers claiming neural networks outperforming tree models on tabular data, and the subsequent series of refutation papers, this topic is timely and important to the community. While the Tabular Data: Deep Learning is Not All You Need paper has many similarities to this paper, this paper builds upon the prior by moving away from the limited and convoluted scope of the datasets used in the original neural network papers and towards a larger and more unified benchmark of 45 tabular datasets. The number of datasets is quite important, as to get a high confidence perspective of the state of these systems, many more datasets are required than are typically contained in papers claiming SOTA with a new method. The method to select datasets is reasonable (although I'd like future work to consider incorporating missing data into datasets as it is very common in real-world datasets).

Beyond an improved benchmark, this paper also covers reasons for the performance gap, which is of practical importance to the development of tabular NN models. In particular the authors make a very nice observation on data rotation which reverses the performance order of the models.

**Weaknesses:**

L247: Random forest feature importance is not a very accurate method for computing importance, and could induce biases when selecting features to prune according to the definition of importance by random forest (RFs may leverage features more closely to GBMs than NNs, causing important features of GBMs to be removed later than important features for NNs). Instead, iterative permutation shuffling feature importance and pruning on a per-model basis is found to be a more effective method that more closely aligns with the true importance. This would also enable a deep dive into which features are important for trees vs NN models. Overall this is a fairly minor point, but wanted to make the authors aware of alternatives. Source: https://explained.ai/rf-importance/

While the authors make the wise decision of setting the epochs to 300 and using early stopping for NN models (with 300 being a large value that often won't be reached), this is oddly not done for tree models. For example, num_estimators is used as part of the search space for XGBoost, Random Forest, and Gradient Boosting. For Random Forest, this is strange because RF will not overfit to validation data, and having fewer trees will simply make the model worse on average. It would be much more sample efficient to fix the num_trees to a single large value such as 2000, as it is redundant to search. For XGBoost and Gradient Boosting, simply use early stopping with a large patience and a very large num_estimators value, such as 10000. This will avoid needing to search for the optimal value and will greatly improve search efficiency. For the paper results, this likely won't change much due to the severity of the gap in performance between NN and trees, but it will be very important for future work if NN models become more competitive.

**Additional Feedback:**

While the authors mention that the AutoMLBenchmark has issues with incorporating synthetic datasets and non-tabular datasets, I find that the codebase may be a powerful tool to re-use as it supports many datasets and has ready to use implementations of many algorithms with heavy adoption within the tabular AutoML community. Re-use and refinement of a singular tabular data benchmarking repo that is scalable and trusted for research in this vein would be greatly beneficial to the community, and it is unclear if the authors benchmark will become yet-another-tabular-benchmark or if they intend to differentiate their benchmark beyond the improvement in the choice of datasets / removal of missing values. I would be interested in clarification from the authors on this point.


**Documentation:**

Yes, the datasets are available on OpenML and model implementations are provided with hyperparameter search spaces for the models in a public GitHub repo.

**Ethics:**

There are no ethics concerns with this paper nor its datasets.

**Relation To Prior Work:**

Yes, the authors cite important prior work and describe how their work extends prior work.

**Summary And Contributions:**

The authors propose a comparison of tree based methods with modern deep learning techniques on small-medium sized Tabular data. They perform a large-scale hyperparameter tuning sweep of numerous tree and deep learning models across a newly proposed benchmark of 45 datasets and compare results of the models. They investigate the reasons why tree models consistently outperform deep learning models including uninformative features, data orientation preservation, and irregular functions.

---

> ### Author Response · Authors · 2022-08-24
> **Answer**
>
> We thank the reviewer for the positive review, and we now answer the questions:
>
> > Random forest feature importance is not a very accurate method for computing importance, and could induce biases when selecting features to prune according to the definition of importance by random forest (RFs may leverage features more closely to GBMs than NNs, causing important features of GBMs to be removed later than important features for NNs). Instead, iterative permutation shuffling feature importance and pruning on a per-model basis is found to be a more effective method that more closely aligns with the true importance.
>
> We agree with the reviewer that this would improve our experiment. We note that if important features for RFs are more similar to important features for GBMs than those for NNs, this would only reinforce the conclusion of Figure 5 (as we remove features in increasing order of RF feature importance, the gap between GBMs and NNs would reduce more quickly if the feature importances we used were more correlated with NNs features’ importance)
>
> > While the authors make the wise decision of setting the epochs to 300 and using early stopping for NN models (with 300 being a large value that often won't be reached), this is oddly not done for tree models. For example, num_estimators is used as part of the search space for XGBoost, Random Forest, and Gradient Boosting. For Random Forest, this is strange because RF will not overfit to validation data, and having fewer trees will simply make the model worse on average. It would be much more sample efficient to fix the num_trees to a single large value such as 2000, as it is redundant to search. For XGBoost and Gradient Boosting, simply use early stopping with a large patience and a very large num_estimators value, such as 10000. This will avoid needing to search for the optimal value and will greatly improve search efficiency. For the paper results, this likely won't change much due to the severity of the gap in performance between NN and trees, but it will be very important for future work if NN models become more competitive.
>
> We thank the reviewer for raising this point. In our desire to be unbiased, we used the hyper-parameters distributions provided by hyperopt-sklearn, which is a popular package for setting hyperparameters for tree-based models. However, we agree with the reviewer that tuning the number of estimators for tree-based models, and especially for random forest, makes little sense, and introduces a difference with our neural networks evaluation. We have therefore changed the hyperparameter spaces of tree-based models to match the reviewer’s advice, and re-run our computations. In the Appendix A.9.1, we compare this new setting to our previous one: the results we obtain in both settings seem relatively similar, and very close for XGboost. Our new setting seems to improve the random search performance for Random Forests a little but worsens it for Gradient Boosting Trees.
>
> > While the authors mention that the AutoMLBenchmark has issues with incorporating synthetic datasets and non-tabular datasets, I find that the codebase may be a powerful tool to re-use as it supports many datasets and has ready to use implementations of many algorithms with heavy adoption within the tabular AutoML community. Re-use and refinement of a singular tabular data benchmarking repo that is scalable and trusted for research in this vein would be greatly beneficial to the community, and it is unclear if the authors benchmark will become yet-another-tabular-benchmark or if they intend to differentiate their benchmark beyond the improvement in the choice of datasets / removal of missing values. I would be interested in clarification from the authors on this point.
>
> Our initial goal was to create an easily accessible standard suite of datasets: we do not know of any tabular benchmark suite with a set of well-controlled inclusion criteria focusing on tabular data. Given that our benchmark is based on OpenML datasets, it is easy to reuse in a flexible way, with other frameworks and tools, without worrying about data processing variations, and guarantees that our benchmark will be usable for a very long time. We think this is useful as there is no agreed upon standard benchmark for tabular data: a first step is to agree on data and procedures (as what imagenet achieved for computer vision).
>
> Our Github repo also provides a framework for people wanting to benchmark their own algorithm, and we hope the community will add more algorithms to it. To this aim, we’re actively working on making our framework easier to run, and we’re thinking about changing our codebase to be based on already existing ones. Our plan was to use https://github.com/benchopt/benchopt, but the AutoMLBenchmark codebase could also be a good idea.

---

> > ### Comment · Reviewer_Damk · 2022-09-02
> > **Response to Authors**
> >
> > I first want to thank the authors for responding to my review.
> >
> > ## Regarding your new section A.9.1 attempting to address my concern on early stopping:
> >
> > [Note: this section is less important than the next section in terms of my review]
> >
> > > In the second setting, we set the number of estimators to a high number for each model, and use early
> > stopping for GradientBoostingTree and XGboost, with a "patience" of 20 and a default tolerance, and
> > a maximum number of 1000 estimators. For RandomForest, we do not use early stopping and set the
> > number of estimators to 250.
> >
> > The claim that random search of iterations improves GBM models compared to early stopping is a claim that I simply cannot believe to be true when early stopping is implemented with reasonable maximum iterations and patience. 1000 estimators is far too few as an upper bound. There is little reason to not set it to 10,000 or even 100,000. 1,000 will often be reached and will not be enough to train (as is evident by your random search space going all the way to 6000). It is thus not surprising that this would make performance worse. Beyond this issue, a patience of 20 is too small for tree models. A patience of 100 would be much more stable on small datasets (I personally use 150+ patience in practice for small datasets).
> >
> > Similarly, 250 trees is too few for random forest. Given you previous trained up to 3000, it seems odd that you chose 250 as the fixed value. 1000 is a good point at which diminishing returns kick in heavily for random forests, however this is mostly my opinion.
> >
> > ## Regarding the benchmark itself:
> >
> > > Our initial goal was to create an easily accessible standard suite of datasets: we do not know of any tabular benchmark suite with a set of well-controlled inclusion criteria focusing on tabular data. Given that our benchmark is based on OpenML datasets, it is easy to reuse in a flexible way, with other frameworks and tools, without worrying about data processing variations, and guarantees that our benchmark will be usable for a very long time. We think this is useful as there is no agreed upon standard benchmark for tabular data: a first step is to agree on data and procedures (as what imagenet achieved for computer vision).
> >
> > I am still confused why AutoMLBenchmark does not fulfill that criteria as a backbone benchmark orchestrator. The concern of inclusion criteria is an entirely separate concern from the actual code implementation of the benchmark, and AutoMLBenchmark allows for [custom benchmark dataset configs to be provided / used](https://github.com/openml/automlbenchmark/blob/master/docs/HOWTO.md#add-a-benchmark), [custom model support](https://github.com/openml/automlbenchmark/blob/master/docs/HOWTO.md#add-an-automl-framework), as well as includes [logic for separation of python environments for actual benchmark framework execution to prevent installation conflicts between packages](https://github.com/openml/automlbenchmark/blob/master/frameworks/RandomForest/setup.sh), and [versioned configs for reproducing past benchmarks](https://github.com/openml/automlbenchmark/blob/master/resources/frameworks_2021Q3.yaml).
> >
> > Meanwhile, the code provided by the authors has no setup.py or requirements.txt files that I can find, nor any installation readme making it unclear how one could easily reproduce an experiment. The instructions merely mention git cloning the repo and using wandb, this is not sufficient to have the dependencies needed to actually run the code / models (for example, nowhere does it install pandas, xgboost, etc.). The config files for hyperparameters for all models are contained in a single file instead of separated by model, making the codebase increasingly unwieldy and requiring new contributors of models to edit pre-existing files instead of providing a cleanly separated logic based on templates.
> >
> > I have pretty significant concerns that the author's claim of usability, reproducibility, and extensibility in their benchmark tooling is overstated, but it does not override the good I see in the rest of the paper.

---

> > > ### Author Response · Authors · 2022-09-03
> > > **Answer**
> > >
> > > We thank the reviewer for engaging with our answer.
> > >
> > > # Early stopping
> > > > The claim that random search of iterations improves GBM models compared to early stopping is a claim that I simply cannot believe to be true when early stopping is implemented with reasonable maximum iterations and patience. [...] 1000 estimators is far too few as an upper bound. There is little reason to not set it to 10,000 or even 100,000. 1,000 will often be reached and will not be enough to train (as is evident by your random search space going all the way to 6000).
> > >
> > > Though the decrease in performance could be partly due to the removal of training data for early-stopping validation, we agree that increasing the maximum number of estimators or the patience should improve performances. We believe that the values suggested by the reviewer correspond to quite high computing budgets compared to what practitioners are using in general for tree-based models. For instance, it is a lot higher than Scikit-learn’s default of 100 estimators, and hyperparameter tuning frameworks use sensibly lower values: auto-sklearn uses 512 estimators and a maximum patience of 20 for its GradientBoosting hyperparameters space, and the hyperopt-sklearn hyperparameters we were using at first has a maximum of 1000 estimators for GradientBoostingTrees (but 6000 for XGBoost, as pointed out by the reviewer).
> > >
> > > That being said, we have checked and the limit of 1000 estimators is indeed often reached. **We have launched new runs with a higher number of maximum estimators and a higher patience, which we will include in the final version.** We stress that using the values suggested by the reviewer should only reinforce our conclusion that tree-based models outperform neural networks on tabular datasets.
> > >
> > >
> > >
> > > > Similarly, 250 trees is too few for random forest. Given you previous trained up to 3000, it seems odd that you chose 250 as the fixed value. 1000 is a good point at which diminishing returns kick in heavily for random forests, however this is mostly my opinion.
> > >
> > > We used 250 trees based on [Oshiro et al, “How Many Trees in a Random Forest?”], which finds a very marginal increase in performance after 128 trees. Furthermore, using a larger number of trees like 1000 would have made our random forests much slower to run. Finally, we note that using 250 trees instead of the default settings from hyperopt-sklearn has improved performance.
> > >
> > > # Codebase
> > > We believe that the most interesting contributions of our study lie in defining a standard set of datasets with strict “tabular” properties, in evaluating new models on this set of datasets, taking into account hyperparameter tuning and sharing our compute-intensive results, and in studying the different inductive biases of neural networks and tree-based models, rather than in the codebase itself. While our codebase can be used to check our results, we agree that it could be made easier to use and extend for practitioners. To this aim, we’re evaluating how we could incorporate our dataset selection and benchmarking procedures into a well-engineered existing project like AutoMLBenchmark, which would provide indeed useful features.
> > >
> > > We have added a requirements.txt file to our repo.

---

### Official Review · Reviewer_FDZb · 2022-07-28
**Review for "Why do tree-based models still outperform deep learning on tabular data?"**

**Rating:** 7
**Confidence:** 5
**Clarity:** The paper is well-written and easy to…

**Strengths:**

The paper defines 45 datasets from various domains with different data sizes and feature types, which is a good contribution to tabular data community. The empirical investigation for inductive biases looks interesting.

**Weaknesses:**

The paper only conducts experiments on small and medium size of datasets, people would wonder the comparison between tree-based model and deep learning based model on large scale datasets. Author may need to add more large datasets, although most large datasets are unable to satisfy the criteria proposed in the paper. Large datasets in the paper are truncated to 50,000 samples, e.g., original Covertype owns nearly 500,000 samples, but the paper only keeps one tenth of original dataset. [1] shows the superior performance of FT-transformer on large dataset, so the author may need to carefully claim "tree-based model still outperform deep learning on small scale tabular data". And large dataset with 50,000 samples defined in this paper still looks like a tiny dataset for me.

[1]. Gorishniy, Yury, et al. "Revisiting deep learning models for tabular data." Advances in Neural Information Processing Systems 34 (2021): 18932-18943.

**Additional Feedback:**

Please see the weakness.

**Correctness:**

The paper constructs small tabular datasets and probably evaluate different methods based on these datasets.

**Documentation:**

Yes, all the references are listed.

**Relation To Prior Work:**

The paper clearly discussed the difference from previous contributions.

**Summary And Contributions:**

The paper conducts comprehensive experiments to demonstrate that tree-based model still outperforms the deep learning based model. To understand this gap, the author also conducts an empirical investigation into the differing inductive biases of tree-based models deep learning model.

---

> ### Author Response · Authors · 2022-08-19
> **Answer**
>
> We thank the reviewer for this positive review, and we now answer your questions:
>
> > The paper only conducts experiments on small and medium size of datasets, people would wonder the comparison between tree-based model and deep learning based model on large scale datasets. Author may need to add more large datasets, although most large datasets are unable to satisfy the criteria proposed in the paper
>
> We agree that a proper comparison of neural networks and tree-based models on large-scale datasets would be very useful and interesting. However, while writing this paper, we have indeed found it hard to gather a sufficient amount of open-source datasets of large scale. We have added a new section to the appendix (A.5.2) investigating which parts of our dataset filtering process remove the largest number of datasets. The conclusion is that gathering a sufficient number of large datasets would require removing a lot of restrictions at once, which would change our benchmarks a lot and could make them harder to interpret. Another solution could be to spend more time looking for open-source large-scale datasets in the wild, but this would be a very time consuming endeavor that we leave for future work. In addition, assuming that the datasets on openml reflect a wider practice, these large settings are less frequent.
>
> > Large datasets in the paper are truncated to 50,000 samples, e.g., original Covertype owns nearly 500,000 samples, but the paper only keeps one tenth of original dataset.
>
> We made this choice to have a consistent benchmark without too much variation in dataset size. This allows us to compare this setting (\~50K samples) to the medium-sized setting (\~10K). Given the small number of large-scale datasets we have, large variations in dataset sizes would make the aggregate results very hard to interpret.
>
> > [1] shows the superior performance of FT-transformer on large dataset, so the author may need to carefully claim "tree-based model still outperform deep learning on small scale tabular data".
>
> We have changed the title to “Why do tree-based models still outperform deep learning on *typical* tabular data?” to emphasize that we haven’t run our benchmarks on all tabular data settings, while stressing the fact that medium-sized datasets are very common.

---

### Official Review · Reviewer_UaaX · 2022-07-28
**Meticulous evaluation to study highly relevant questions**

**Rating:** 8
**Confidence:** 4

**Strengths:**

The authors go beyond previous benchmarks in several commendable ways.
- Clearly there is a consciousness of the importance and variation (and arbitrariness) of hyperparameter choices in models. Wonderful to see the hyperparameter grid be part of the benchmark. Hope this sets a standard; the non-smooth and small nature of tabular data problems motivates this concern very well in practice.
- "Not too easy" processing (line 115) is very sensible and I've seen it done in paper evaluations, but somehow not on a benchmark of this type - great!
- The tabular datasets themselves are larger and more varied than prior work represents.
- Confidence bars on everything, and the data behind them also provided as part of the benchmark. Given the size of the final error bars, it's clear the scrupulousness is necessary to draw conclusions from the data.

Meticulous benchmark papers for tabular data are not very common, and certainly underrepresented relative to their usefulness and need.

**Weaknesses:**

The work delivers some high-level takeaway messages that would be unsurprising to practitioners already working with tabular data:
- Tree-based learners tend to still be superior to deep learners on tabular data.
- Tree-based learners have an easier time handling nonsmooth or uninformative features.
The findings are unsurprising because they're widely understood (though this benchmark is now one of the easiest ways I know of to demonstrate these findings).

Another area of improvement for the benchmark is in testing other uses of these models beyond 0-1 classification:
- I'm a little nervous about preprocessing the data to make binary classification problems out of multiclass problems. The geometry of these may result in different characteristics to multiclass problems, and there are plenty of datasets for which the paper's method (using just the two most common labels) discards a lot of the interesting signal.
- It would be nice to evaluate using many different metrics instead of just accuracy. For confidence-rated prediction, class-imbalanced test data distributions, and other challenging scenarios, different loss functions suggest themselves. There's something to be learned from evaluations in the past, like [Caruana & Niculescu-Mizil '06 "An empirical comparison of supervised learning algorithms"] which justify doing this. This also provides a perspective on the different losses used to train these models.
- The method used in Sec 3.4 still could have has issues in aggregating results, as it would inherit selection biases in overrepresenting particular datasets. Possibly relevant is a game-theoretic take on evaluating performance on datasets, e.g. from Balduzzi et al. '18 "Re-evaluating Evaluation."

The problem space benchmarked is not very large (7 models * 45 datasets) because the hyperparameter space search is meticulous. The datasets and model configurations are so extensively documented, and this is a problem of significance to many people. It would be helpful to crowdsource expansion of the grid search, and of the benchmark in general to new models/datasets. The authors have set the standard and framework for evaluation; with a small amount of continuing effort, this could be a more long-lasting resource to the community.

**Additional Feedback:**



**Clarity:**

The paper is very clear. Both results in Sec. 4.2 are practical findings that could be surprising to many tabular data learners, and perhaps could be moved up in the paper; on inspection they're currently not mentioned before/except in this section, in the middle of page 5.

This paper's benchmarking of hyperparameter variation with respect to a collection of datasets would be of interest to explore the properties of the learned embedding. When deciding whether to explore deep learning on a particular tabular problem, a major motivator is getting a learned embedding of the data for downstream tasks. It would be interesting to see the performance of the embedding method at preserving similarities between examples, how well nonlinear classifiers do in the learned space, and how this all relates to test set accuracy.

**Correctness:**

The method for doing evaluations seems sound. Everything in the github repo is posted as advertised.

**Documentation:**

The combination of the repo (for the models) and OpenML (for the data) adequately satisfy documentation needs.

**Ethics:**

From OpenML datasets; any discussion should go to there.

**Relation To Prior Work:**

Clearly discussed. Would be good to cite Caruana & Niculescu-Mizil '06 and borrow from there.

**Summary And Contributions:**

"Tabular" data comprises most of the uses of machine learning in the wild, including the large majority that don't get press; ensembles of tree-based learners, e.g. with boosting/generalized additive models, have long predominated in this area. Yet deep learning models have shown distinct advantages for more structured data. This paper establishes a benchmark of 45 varied tabular datasets to assess the situation, as the title question asks.

The authors benchmark state-of-the-art tree-based learners against state-of-the-art deep architectures for these datasets, finding that the tree-based learners perform significantly better. Special focus is put on accounting for different hyperparameter optimization budgets. Investigating why the tree-based learners perform better by transforming training and test data in various ways, the authors highlight the importance of dealing with uninformative features, rotation non-invariance, and non-smooth prediction functions.

---

> ### Author Response · Authors · 2022-08-24
> **Answer**
>
> We thank the reviewer for the positive review, and we now answer the questions:
>
> ## Various questions / suggestions
>
> > The work delivers some high-level takeaway messages that would be unsurprising to practitioners already working with tabular data:
>
> We agree that practitioners are often aware of these facts, but after numerous discussions, we think this is not always the case in academia, which is understandable as many recent papers claim to outperform tree-based models. Hence we felt the need for common benchmarks, solid evidence, and hypotheses on the reasons of the differences.
>
> > Another area of improvement for the benchmark is in testing other uses of these models beyond 0-1 classification:
>
> Adding multi-class classification to our benchmark would be an improvement, and we will suggest it for future work.
>
> > I'm a little nervous about preprocessing the data to make binary classification problems out of multiclass problems. The geometry of these may result in different characteristics to multiclass problems, and there are plenty of datasets for which the paper's method (using just the two most common labels) discards a lot of the interesting signal.
>
> While it is true that binarizing multiclass dataset can change the geometry of the problem a lot, and discard interesting signal, we think that the resulting dataset, while different from the original one, constitutes an interesting dataset with clear tabular data characteristics. Furthermore, we want to stress that we check the difficulty of the dataset *after* binarisation, to make sure we haven’t discarded all signal.
>
> > Both results in Sec. 4.2 are practical findings that could be surprising to many tabular data learners, and perhaps could be moved up in the paper; on inspection they're currently not mentioned before/except in this section, in the middle of page 5.
>
> We have added a summary of these findings in the introduction.
>
> ## Using different metrics
>
> We thank the reviewer for mentioning the [Caruana & Niculescu-Mizil '06] paper which we had missed. We’ve included it in the “Related work” section. We agree that adding more metrics to our benchmark would make it more interesting, but we think that for the tasks we chose (balanced 2-classes classification and regression), the simple metric we chose captures a sizable part of the story.
>
> [Caruana & Niculescu-Mizil '06] stress that classifiers come with confidence scores that are themselves thresholded to become classes, and that calibration, quantifying how much the classifier scores reflect error probabilities, is also important. This is an important point. Yet, it is probably a second-order analysis compared to the main message of our work, and we believe that calibration, including post-hoc calibration of the classifiers, would not change the overall ranking (as is the case in [Caruana & Niculescu-Mizil '06]) . Nevertheless, a quantification of calibration error would be a welcome complementary measure. We cannot however report it without re-running the complete set of benchmarks, as we have not stored the individual predictions, and thus we will mention this as follow-up work.
>
> ## Aggregating results and game-theoretic metrics
>
> We thank the reviewer for mentioning this paper, which we found very insightful. We decided to stick with our aggregation method for several reasons:
> - It is simple to understand, and thus to communicate to practitioners and other researchers
> - Relatedly, it is similar to previous works, for instance [Matthias Feurer et al. Auto-Sklearn 2.0: Hands-free AutoML via Meta-Learning, 2021.], which allows for easy comparisons. We add that our aggregation method is very similar to that of the other article mentioned by the reviewer ([Caruana & Niculescu-Mizil '06 "An empirical comparison of supervised learning algorithms"]), the main difference being that the “zero score” is defined to be the score of the 10% worse algorithm in our case, while it correspond to the score of an artificial baseline (which predict the most common class) in the latter article.
> - As we have a strict criterion on dataset difficulty, we don’t have many easy tasks that would bias our results. Similarly, we don’t include in our benchmark algorithms which are not competitive.
> - In our case, the number of tasks is small enough that we can check that the aggregated results are faithful to the per-datasets results. This is what we do in Appendix A.3, where we present the unnormalized results for each dataset.
>
> That being said, we might consider using game-theoretic metrics if we were to extend our benchmark to a larger number of more diverse algorithms and tasks.
>
> ### Evaluating the usefulness of the learnt embeddings for downstream tasks
>
> We agree with the reviewer that this would be useful, but we think this goes beyond the scope of the present paper and will suggest it for follow up work.

---

### Official Review · Reviewer_gCXf · 2022-07-29
**Review for 'Why do tree-based models still outperform deep learning on tabular data?'**

**Rating:** 6
**Confidence:** 3

**Strengths:**

- The paper is clearly written and well-motivated. The problem considered in the paper is an important one to enable the adoption of NNs in domains apart from vision and NLP.
- The experiments in the paper are interesting and well-designed. The considered inductive biases make intuitive sense as the ones likely to account for the difference in performance between NNs and tree-based models.
- Some of the conclusions in the paper, such as the fact that breaking invariances can lead to better performance on tabular data, are potentially critical to the design of more performant NN architectures for tabular data

**Weaknesses:**

- One area concern for me in the experimental design was the presence of several magic numbers and vague descriptions of criteria for the choice of reference tabular datasets in Section 3.1. For example, why are high-dimensional datasets not used? What aspect of the dimensionality would invalidate the results in the paper? Why is 3000 the cutoff for 'too few samples'? Similar concerns impact the choices in Section 3.2.
- Other types of data smoothing apart from Gaussian smoothing could be considered.
- The paper could have included some experiments with NNs that break rotational invariances.

**Additional Feedback:**

See weaknesses

**Clarity:**

Apart from some clarity issues in the experimental design description, the paper is well-written.

**Correctness:**

The experimental design is largely sound, but has some choices that need to be justified better for soundness' sake.

**Documentation:**

The benchmark links to well-documented code.

**Ethics:**

None.

**Relation To Prior Work:**

I am not well-versed enough in the prior work to judge this, but the paper seemed to contain sufficient references.

**Summary And Contributions:**

This paper aims to understand and establish, based on empirical evidence, if and why tree-based models tend to perform better than neural networks on tabular data. Existing work has indicated that this is the case, and the current paper aims to conclusively determine this over a wide range of tabular datasets. The first preliminary result, shows that the average performance of tree-based models is better than NN-based models, when controlling for hyperparameter search via the number of random searches used. The paper then aims to understand why this is the case, by considering the following 3 case feature-level hypotheses: i) neural networks perform better on smoothed-out features; ii) they are less robust to the presence of uninformative features; iii) they are rotationally invariant even when the data is not. The paper puts forth these as the main inductive biases that lead to the superior performance of tree-based models.

---

> ### Author Response · Authors · 2022-08-19
> **Answer**
>
> We thank the reviewer for this positive review, and we now answer the questions:
>
> ## Discussion on the experimental design
>
> We agree with the reviewer that certain cut-offs had to be chosen quite arbitrarily, as we tried to strike a balance between having numerous datasets and having a consistent benchmark, which allows us to answer a specific question (do tree-based models still outperform neural network on the “typical” kind of tabular data?) and to delve into the performance gap to study the inductive biases of different algorithms.
>
> We understand that arbitrary numbers may be cause for concern, but in this case we think this is not an issue. Indeed:
>  - we set all these numbers *before* running the benchmark, to be completely unbiased.
> - The set of datasets chosen for our benchmark is not very sensitive to the specific cut-offs we chose, as most datasets are not close to the cutoff points. For instance, changing the minimum number of samples from 3000 to 5000 only removes 5 datasets among 45, and changing the minimum number of features from 4 to 6 only removes 4 datasets.
>
> | Num samples min     | Num features min |Num dataset selected|
> | ----------- | ----------- |----------- |
> | 3000      | 3       |45|
> | 5000   | 3       |40 |
> | 3000      | 4       |45|
> | 5000   | 4       |40 |
> | 3000   | 5       |41 |
> | 3000   | 5       |37 |
>
> We would love to know more about the performance of neural networks and tree-based models without the restrictions we imposed (no missing data, no high-dimensional datasets), but we think that removing each one would require a meticulous comparison of its own, for instance with different baselines to compare to.
>
> Another benefit of restrictions like removing high-dimensional datasets or keeping only hard enough datasets is to make the benchmark faster to run for other researchers wanting to use them or build on our work.
>
> We have changed the title to “Why do tree-based models still outperform deep learning on *typical* tabular data?” to emphasize that we haven’t run our benchmarks on all tabular data settings.
>
> ## Experiments with neural networks not invariant to rotation
>
> We thank the reviewer for raising this point. Actually, as for standard vision transformers, the FT Transformers we use are not invariant by rotation because there is an initial point-wise step for the “Feature Tokenizer”. (We agree with the reviewer that the rest of the FT Transformer would be rotationally invariant). Figure 6  indeed shows that random rotations decrease FT Transformer’s accuracy. We think this non-invariance is related to the fact that the FT Transformer is much more robust to uninformative features than MLP-like neural networks (Figure 5). We have emphasized this in Section 5.4. An interesting albeit more speculative point is that empirically, the FT Transformer does seem less sensitive to the rotation of the features than tree-based models, and it could be useful to look for neural networks architectures which are as sensitive to the rotation of the features as tree-based models.
>
>
> ## Other types of smoothing
>
> We agree with the reviewer that a more broad set of experiments could have been considered; however we think a Gaussian smoothing is the most adapted to back our hypothesis (“Neural networks struggle to learn irregular patterns”). A KNN-smoothing, for instance, would have introduced other biases for points without close neighbors.

---

### Author Response · Authors · 2022-08-29
**Summary of the discussion**

We thank all the reviewers for their valuable comments and remarks, which helped improve our current submission. We’d like to emphasize that most of the reviewers have considered our contribution good and significant, and we thank them for this positive feedback.
In this comment, we would like to summarize the main criticisms, answers, and changes to the paper during the review process.

## Discussions on our contribution
### Clarification on our contributions (w58A)

Our contribution is threefold: 1) we define a standard set of datasets with strict “tabular” properties 2) we evaluate new models on this set of datasets, taking into account hyperparameter tuning and sharing our compute-intensive results 3) we delve into the performance gap between tree-based models and neural networks, and shed light on their different inductive biases. The third contribution is only possible because of the clear-cut definition of a benchmark suite. Indeed, an heterogeneous benchmark would have created new sources of performance difference, washing out the signal of the empirical study.

### Homogeneity of our benchmark (w58A, FDZb)

Reviewer w58A wonders if our benchmark is too homogeneous, and reviewer FDZb notices that we have truncated some large datasets to make the size of our datasets more homogeneous. We have indeed strived to build a homogeneous set of datasets! This allows us to interpret our benchmark as measuring performance on typical tabular datasets, and to leverage our benchmark to study inductive biases well-suited to tabular data. Though our benchmark does not include some crucial challenges for tabular data (such as missing data), performing well on our benchmark is probably a first step to have a good algorithm for tabular data. While new approaches should also be evaluated in more challenging conditions, we think that it might be best to separately develop inductive biases well-suited for tabular data in the strictest sense (i.e well-suited for our benchmark), and methods made to deal with missing data or high-cardinality variables.

We have changed the title to “Why do tree-based models still outperform deep learning on *typical* tabular data?” to emphasize that we haven’t run our benchmarks on all tabular data settings.

### Comparison with previous benchmarks (w58A, rCUL)

Reviewer rCUL asked for more details on the difference between our benchmark and previous ones, and w58A questioned the usefulness of creating a new set of datasets when others already exist. We have added a new section in the Appendix (A.6) comparing our benchmark to previous ones (CC-18 and the AutoML benchmark) and quantifying the overlap, which is surprisingly small. In this section, we also detail the advantages of our benchmark which make it useful: our benchmark is clearly about tabular data, is more homogeneous, does not contain datasets which would not contribute meaningfully to the comparison, contains both classification and regression tasks, and still contains 45 datasets despite having stricter inclusion criteria (more details in A.6).

## Requested changes and additions

### Change of hyperparameter space (DamK)

We have changed the hyperparameter spaces of tree-based models to match the reviewer’s advice (to not optimize the number of estimators), and re-run our computations. In the Appendix A.9.1, we compare this new setting to our previous one.

### Comparison of our random search procedure with Bayesian optimisation (w58A)
We have added a new section in the Appendix A.7, where we try Bayesian optimization, and find that it doesn’t change our conclusion.
### Statistical analysis of our experiments (w58A)
Added to the Appendix (A.4)
### More large datasets (FDZb, rCUL)
Reviewers FDZb and rCUL would have liked more results on large datasets, which we agree would be very interesting. We have added a new section to the appendix (A.5.2) investigating our dataset filtering process. The conclusion is that gathering a sufficient number of large datasets would require removing a lot of restrictions at once, which would change our benchmarks a lot and could make them harder to interpret. Another solution could be to spend more time looking for open-source large-scale datasets in the wild, but this would be a very time consuming endeavor that we leave for future work. In addition, assuming that the datasets on openml reflect a wider practice, these large settings are less frequent.
### Other additions
Because they were beyond the scope of the paper, or sometimes because we simply lacked time, some interesting suggestions could not be added, and we have suggested most of them for future works. This includes adding multi-class classifications problems, studying the usefulness of learnt embeddings of neural networks for downstream tasks, using more diverse metrics, using different methods for score aggregation, small changes to our experiments, and extending our benchmark to other settings (small datasets, missing data etc.).

---

### Meta-Review · Area_Chair_QCD4 · 2022-09-08

**Recommendation:** Accept
**Confidence:** 4

**Metareview:**

The paper proposes a benchmark comparing modern neural network architectures with gradient boosting.

Overall the paper was reviewed quite positively with an average score of 6.33.

Most criticism by the reviewers was addressed by the authors.

Major remaining issues are:

1) While the results of BO vs random search indicated by reviewer w58A is indeed surprising and worth further investigation.

2) That the uncertainty estimates are only sampled by shuffling a single random search run is indeed not ideal. This makes the uncertainty in the later parts of the plots less reliable.

3) The benchmarks makes no use of a standardized platform like AMLB. While I would very much prefer this, I don't think it can be a requirement for such a paper.

4) The mismatch between early-stopping and tuning the number of iterations of NNets and Boosting.


While all these points are valid, I would argue that the main findings of the benchmark are correct, the benchmark set itself is interesting and further research can look into some of the issues mentioned in 1) - 4).

---

### Decision · Program_Chairs · 2022-09-16

Accept